# Deep-Learning-Based Defective Bean Inspection with GAN-Structured Automated Labeled Data Augmentation in Coffee Industry

Yung-Chien Chou [1], Cheng-Ju Kuo [1], Tzu-Ting Chen [1], Gwo-Jiun Horng [2],
Mao-Yuan Pai [3,*], Mu-En Wu [4], Yu-Chuan Lin [1], Min-Hsiung Hung [5], Wei-Tsung Su [6],
Yi-Chung Chen [7], Ding-Chau Wang [2] and Chao-Chun Chen [1,*]

[1] Institute of Manufacturing Information and Systems, Department of Computer Science & Information Engineering, National Cheng Kung University, Tainan 701, Taiwan; tach40125@gmail.com (Y.-C.C.); P96074090@mail.ncku.edu.tw (C.-J.K.); P96084126@mail.ncku.edu.tw (T.-T.C.); duke@imrc.ncku.edu.tw (Y.-C.L.)

[2] Department of Computer Science & Information Engineering, Department of Management Information System, Southern Taiwan University of Science and Technology, Tainan 710, Taiwan; grojium@gmail.com (G.-J.H.); dcwang@stust.edu.tw (D.-C.W.)

[3] General Research Service Center, National Pingtung University of Science and Technology, Pingtung 912, Taiwan

[4] Department of Information & Financial Management, National Taipei University of Technology, Taipei 106, Taiwan; mnwu@ntut.edu.tw

[5] Department of Computer Science & Information Engineering, Chinese Culture University, Taipei 111, Taiwan; hmx4@faculty.pccu.edu.tw

[6] Department of Computer Science & Information Engineering, Aletheia University, New Taipei 251, Taiwan; suwt@au.edu.tw

[7] Department of Industrial Engineering & Management, National Yunlin University of Science and Technology, Yunlin 640, Taiwan; mitsukoshi901@gmail.com

* Correspondence: mypai@mail.npust.edu.tw (M.-Y.P.); chaochun@mail.ncku.edu.tw (C.-C.C.)

**Abstract:** In the production process from green beans to coffee bean packages, the defective bean removal (or in short, defect removal) is one of most labor-consuming stages, and many companies investigate the automation of this stage for minimizing human efforts. In this paper, we propose a deep-learning-based defective bean inspection scheme (DL-DBIS), together with a GAN (generative-adversarial network)-structured automated labeled data augmentation method (GALDAM) for enhancing the proposed scheme, so that the automation degree of bean removal with robotic arms can be further improved for coffee industries. The proposed scheme is aimed at providing an effective model to a deep-learning-based object detection module for accurately identifying defects among dense beans. The proposed GALDAM can be used to greatly reduce labor costs, since the data labeling is the most labor-intensive work in this sort of solutions. Our proposed scheme brings two main impacts to intelligent agriculture. First, our proposed scheme is can be easily adopted by industries as human effort in labeling coffee beans are minimized. The users can easily customize their own defective bean model without spending a great amount of time on labeling small and dense objects. Second, our scheme can inspect all classes of defective beans categorized by the SCAA (Specialty Coffee Association of America) at the same time and can be easily extended if more classes of defective beans are added. These two advantages increase the degree of automation in the coffee industry. The prototype of the proposed scheme was developed for studying integrated tests. Testing results of a case study reveal that the proposed scheme can efficiently and effectively generate models for identifying defective beans with accuracy and precision values up to 80%.

**Keywords:** automatic defect inspection; machine learning; smart agriculture; automation engineering; data augmentation; applied artificial intelligence; GAN optimizer

---

## 1. Introduction

Advancement of recent information technologies evolves the agriculture and food industries. Enterprises are eager to create smart information systems to increase their business competition. Coffee is the largest amount of raw-food material in the world-wide agricultural trade [1]. For example, coffee beans worth over 25 billion USD per year are imported to the Taiwan market. Figure 1 illustrates the production process from green beans to coffee bean packages. The process illustrates from coffee fruit, green beans, defective bean removal, roast beans, to the final bean package. The classes of defective beans are categorized by the SCAA (Specialty Coffee Association of America). Among these stages, defective bean removal (or in short, defect removal) is one of most labor-consuming stages, and many companies investigate the automation of this stage for minimizing human efforts. In addition, the defective bean removal is a critical stage that affects the bean's value, as many experts explicitly point out that defective beans are a key factor for providing high-quality coffee. Thus, removing defective beans becomes a necessary step before brewing for significantly increasing their competition and profits [2].

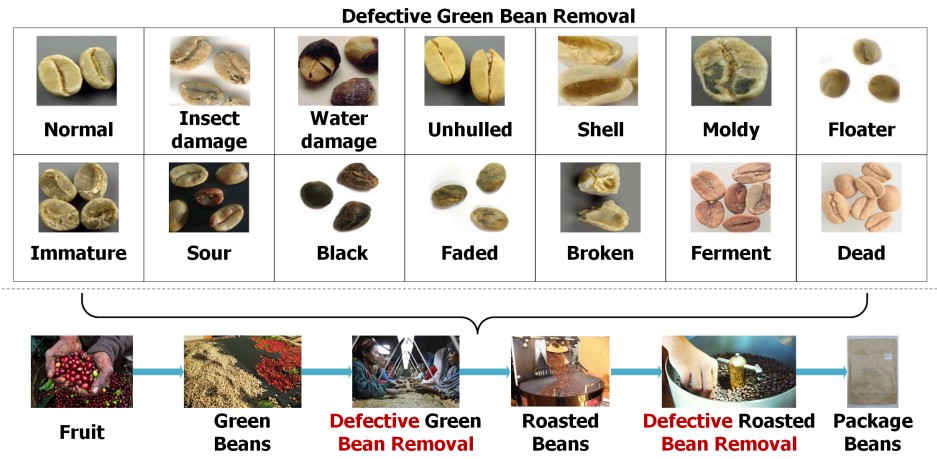

**Figure 1.** An illustration of production process from green beans to coffee bean packages.

The SCAA has classified the defective beans into 13 classes [3], as shown in upper portion of Figure 1. Most popular defective bean removal processes are achieved by manual or mechanical manners in the past decades [1,2,4,5]. Nevertheless, these solutions focus on removing few sorts of defects in the SCAA's classification. If all defects need to be removed completely, the cost of purchasing these products is too expensive. With the advance of robotic techniques, all defective beans can be completely picked off with only one arm equipment, which greatly reduces working space and financial cost. By surveying related literature [6–11], most recent works solve this issue with the vision technologies with artificial intelligence. The difficulties of identifying defective beans in real-world working spaces includes bean overlapping, different surface sizes of beans, different degrees of defective appearances, etc. Turi et al. [12] employed an artificial neural network (ANN) to automatically categorize the coffee beans of four selected locations according to their provenance. They considered certain properties (color, morphology, texture, and a combination of morphology and color features), which cover only some of defective beans. For providing a uniform solution to industries, computational intelligent technologies, such as pattern mining [13–15] or machine/deep learning [3,16–19], bring considerable techniques to develop actionable analytics and prediction to

completely detect all sorts of detective bean patterns at the same time. Deep-learning-based models are inherently more suitable for accomplishing complex tasks with enormous data inputs than generic data-analytic-based models. However, none of them consider the data labeling cost in their solutions, so there is a great barrier to lead these methods to industries. This motivated this work to investigate the issue of identifying all defective beans with less labor effort.

Traditional data labeling over beans may be performed in three ways [3,16,18]. The first one is "one-by-one labeling", in which the user takes pictures for an individual bean and labels the bean for these pictures. This make a coffee always in a specific position or in a blank background, which lead the imprecision inspection in the practice due to the high variance of the trained model. The second one is "all-in-one labeling", in which the user takes a picture of the sufficient amount of beans in a container (e.g., a plate) and labels these beans one by one on a screen. This shall perform better than the first way since it reduces the high variance issue. The third way is "batch labeling", in which the user takes a picture of a lesser amount of beans in a container and labels beans. The process repeats multiple iterations for labeling sufficient amount of beans. The third way is less load than the second one from the human aspect, since the user labels less beans in each iteration. Note that a great amount of labeling time is needed in all these three ways, and this is one of industrial difficulties to lead the learning-based solutions into the coffee industry. We previously have developed a robotic arm guidance method for precisely picking of specified beans [20]. Thus, an efficient defect inspection scheme is needed for adding the intelligent automation capacity to such robotic arms.

In this paper, we propose a deep-learning-based defective bean inspection scheme (DL-DBIS), together with a GAN (generative adversarial network)-structured automated labeled data augmentation method (GALDAM) for enhancing the proposed scheme, so that the automation degree of bean removal with robotic arms can be further improved for coffee industries. The proposed scheme aiming at providing an effective model to a deep-learning-based object detection module for accurately identifying defects among dense beans. The proposed GAN-structured automated labeled data augmentation method is used to greatly reduce labor costs, since the data labeling is the most labor-intensive work in this sort of solutions [7]; the bean labeling is particularly difficult since experts need to label beans on the small and dense bean images, even a great number of bean images, which may incur longer time than generating models in computers. The key idea of our method on greatly reducing labor costs is to iteratively generate new training images containing defective beans in various locations by using a generative adversarial network framework, and these images incur a low successful detection rate meaning that they are not well identified in the current model so that these augmented images are useful for improving model quality. Our proposed scheme requires only a small amount of time for human labeling, which implies that the proposed solution quite satisfies industrial requirements. In summary, our proposed scheme considers aspects of convenience, training-awareness, and effectiveness at the same time during the model generation process.

This work brings two main impacts to intelligent agriculture industries. First, our proposed scheme can easily be adopted by industries as human efforts in labeling coffee beans are minimized. The users can easily customize their own defective bean model without spending a great amount of time on labeling small and dense objects. Second, our scheme can inspect all classes of defective beans categorized by the SCAA at the same time. The above two advantages increase the degree of automation to the coffee industries. Finally, we implement the prototype of the proposed scheme, and apply the prototype to a robotic arm system for conducting integrated tests. Testing results from various performance metrices all reveal that the proposed scheme can efficiently and effectively create superior models for inspecting defective beans.

The rest of this paper is organized as follows. Section 2 introduces some related backgrounds. Section 3 presents the architecture of our proposed deep-learning-based defective bean inspection scheme. Then, Section 4 describes designs of used deep networks and related optimizers. Section 5 discusses the model quality control in the proposed scheme. The case study is shown in Section 6. Finally, we conclude this paper in Section 7.

## 2. Related Work

### 2.1. Survey of Deep Learning Technologies

McCulloch and Pitts [21] pioneered the research of neural network research in 1943, where it was first designed in a mathematical model to simulate the mode of operation of neurons. Based on the computational representation, the first perception models of the adaptive linear neuron were designed in 1957 and 1962, respectively [22]. These solid fundamental achievements brought out more advanced and deep neural networks, such as back propagation neural network, convolutional neural networks (CNN), and long short-term memory (LSTM) using in recurrent neural networks (RNNs) [23]. Recently, YOLO (You Only Look Once) [24] was developed by using a single neural network to directly predict object boundaries and probability of the species, process end-to-end object detection, and treat the object detection as a regression problem. The whole detection method is regarded as a network. With YOLO, the probability of borders and types are predicted from the entire image in only one pass, which is why YOLO is the most popular real-time object detection techniques in image processing applications.

The generative adversarial networks (GANs) was proposed by Goodfellow et al. in 2014 [25], which was designed to train models of generated parameters for deep networks. GANs [25,26] has achieved remarkable results in image generation [27,28], image editing [29] and feature learning [28,30,31]. GANs has been proven to produce high quality images [25,27] and led to the rise of deep learning in computer vision. The key to GANs' success is the concept of confrontational loss, which can be used to generate images that are hard to be distinguished from the real ones by human experts. GANs are widely applied in many fields. Tang et al. [32] proposed a robust and accurate lung segmentor based on a criss-cross attention network and a customized radiorealistic abnormalities generation technique for data augmentation. Nie et al. [33] proposed a supervised GAN framework to synthesize medical images. The network includes a generator for estimating the computed tomography and a discriminator for distinguishing the real computed tomography from the generated ones. Tang et al. [34] proposed a task-driven, discriminatively trained, cycle-consistent generative adversarial network, termed TUNANet to preserve low-level details, high-level semantic information, and mid-level feature representation during the image-to-image translation process, to favor the target disease recognition task.

Recently, ADAM (adaptive moment estimation) [35] has been widely used in deep learning applications, and it is an extension of Stochastic gradient descent (SGD). ADAM can replace the first-order optimization of the traditional SGD and also can iterate calculation and update neural network weights based on training data. This algorithm is easily used in experiments. In addition, ADAM has high computational efficiency and small storage space requirement. ADAM has other advantages, such as its invariant to diagonal rescaling of the gradients. In this work, ADAM is adopted as the underlying optimization tool to deep networks used in this paper.

### 2.2. Generating Deep-Learning Models for Defective Coffee Bean Inspection

Supervised deep learning networks generate qualified models for achieve specified missions (e.g., classification, regression, etc.) with a labeled data set. A standard model generation flowchart for supervised learning applications is given in Figure 2. The first step is to generate the data set for training and testing. For example, we photograph some beans and label defect classes for them in each image. Note that the number of images is usually proportional to quality of generated deep learning models; however, preparing these images consumes huge amount of labor effort. The second step is to compose the training and testing data sets. Usually, 70% of labeling images are randomly selected as the training data set, and the other 30% as the testing data set. Then, in the third step, the training data set is used in generating a model by training the specified deep learning network with reducing the loss function to be acceptable small. Next, in the fourth step, the testing data set is used to validate that the generated model performs well in most cases, instead of incurring the overfitting phenomenon.

If the generated model fails the validation, then the user may prepare more labeled data (i.e., bean images) and repeat the whole procedure until a qualified model is generated.

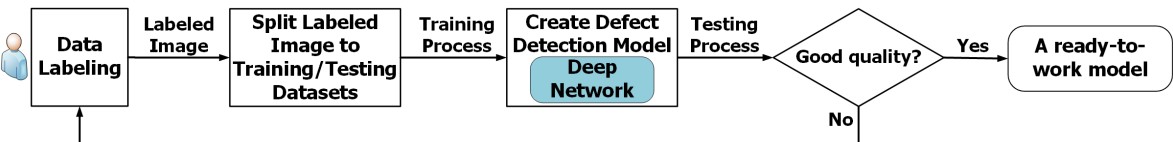

**Figure 2.** A standard model generation flowchart in solutions of the deep learning-based methodology.

In general, any object detection deep network can be trained with this framework to achieve the defect inspection. The YOLO network [24] was adopted as the underlying deep network structure in this work.

## 3. Proposed Deep-Learning-Based Defective Bean Inspection Scheme (DL-DBIS)

Figure 3 shows an illustration of applying defective bean inspection (DBI) model to remove defective beans with a robotic arm [20]. A defective bean picking system (DBPS) implemented in a single-chip computer inspects defective beans in the coffee tray with the DBI deep network and maintains locations of the defective beans in the defective bean location maintainer. Then, the DBPS guides the robotic arm to the top of each defective bean and removes them in sequence, which are performed by the arm routing controller and the target bean removal controller. In this system, a defective bean inspection model generation scheme is needed to obtain a DBI model for the DBI deep network, as shown in Figure 3, so that the defective bean picking system can successfully remove defective beans.

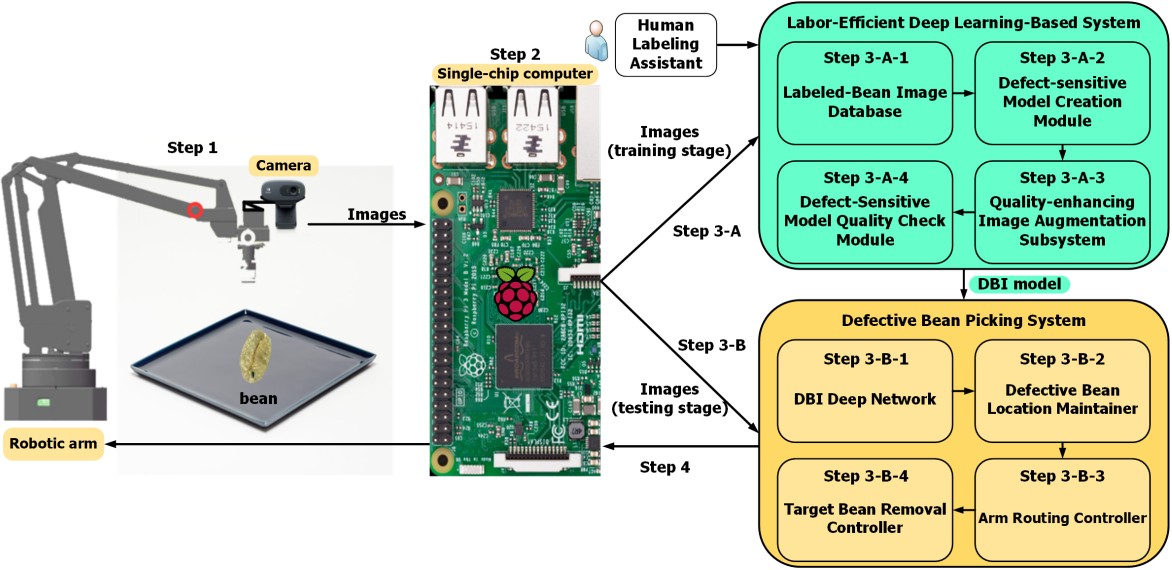

**Figure 3.** Illustration of defective bean removal with a robotic arm.

DL-DBIS includes two operations: one is the model creation and the other is defective bean removal. The model creation operation is to create a DBI model for further inspection, and its operational flow is presented as follows:

**Step 1:** Use a robotic arm equipped with a camera to capture bean images.
**Step 2:** These images, divided into training images and testing images, are transmitted to the computer through the single-chip computer.

Notice that DL-DBIS requires computation resources to achieve the defective bean identification and to inform the robot controller with the communication function. Currently, many productions are

invented and can be adopted. Raspberry Pi, shown in Figure 3, is one of existing single-chip computer productions found in the today's markets and is used in our implementation.

In DL-DBIS, the labor-efficient deep learning-based system is designed to train a proper DBI model to determine whether a bean is a defect or not with Human Labeling Assistant in the training stage. The process is as follows:

**Step 3-A-1:** On receiving training image data, a Labeled-Bean Image Database is used to store these bean images, including bean labels.

**Step 3-A-2:** Then, the Defect-sensitive Model Creation Module creates a defect-sensitive model based on the GAN technique.

**Step 3-A-3:** The Quality-enhancing Image Augmentation Subsystem is used to automatically generate new bean images for training better defective bean inspection models.

**Step 3-A-4:** The Defect-Sensitive Model Quality Check Module checks quality of the defect-sensitive model. If quality is good enough, output the DBI model.

Once a DBI model is created, the Defective Bean Picking System is then used to remove defective beans. The process is described as follows:

**Step 3-B-1:** When receiving the test image data, DBPS uses the DBI deep network to check the beans in the tray.

**Step 3-B-2:** Once the defective beans in the tray are detected, the Defective Bean Location Maintainer gets the position of the coffee beans.

**Step 3-B-3:** The Arm Routing Controller establishes routing and control of the Robotic arm for removing the defective beans.

**Step 3-B-4:** Finally, the Target Bean Removal Controller is of the final step in the Defective Bean Picking System, which has the ability to remove defective beans.

**Step 4:** After implementation of the Defective Bean Picking System is complete, the Target Bean Removal Controller transmits commands to the Single-chip computer by network and controls the robotic arm to remove the defective beans.

*GAN-Based Automated Labeled Data Augmentation Method (GALDAM)*

The GAN (generative adversarial network)-based Automated Labeled Data Augmentation Method (GALDAM) is designed to train a proper model with only a few amounts of human-labeling images (e.g., 20 images in our case study). The GALDAM is the heart of the labor-efficient deep learning-based system shown in Figure 3. Assume a human-labeling training bean image contains each class of defective beans (14 classes in this work), placed sparsely for easily labeling. Thus, only limited data labeling effort is required. The idea of the GALDAM is to augment sufficient amount of training images by using the limited human-labeling bean images with the GAN methodology. To ensure effectiveness of defect inspection capacity, GALDAM also checks whether the generated model is qualified for defective bean inspection. The overall architecture is presented in this section, and related key mechanisms are described in the next sections.

Figure 4 shows the architecture and workflow of the proposed GAN-based automated labeled data augmentation method (GALDAM) based on the above-mentioned design philosophy. Assume a user labels only a few amounts of sparse defective bean images, each of which includes one defective class shown in Figure 1. Thus, all defect classes can be collected (label 0 for normal beans, labels 1~13 for different kinds of defective beans). Implementing the proposed GALDAM has to include six main modules. The first is *labeled-bean image database (LBIDB)*, which stores the labeled bean images for training the optimal model of the defect-sensitive inspection deep network (DIDN). The DIDN is the key to the defect-sensitive model creation (DSMC) module and the detailed structure will be discussed in details in Section 4.1. The second is *DSMC module*, which is designed for creating a defect-sensitive model for the DIDN with inputs, including the human-labeling images ($IMG^{HL}$ and $L^{HL}$) and the augmented images. The third is *GAN-based image-augmentation model*

*generation (GIMG) module*, which is designed for creating a generative network model for a neural network called the bean-shifting deep network (BSDN). The fourth is *low-detection-rate bean image generation (LBIG) module*, which is designed for generating the augmented labeled defective bean image set. The fifth is *augmented-image qualify check (AIQC) module*, which is designed for verifying whether the augmented labeled bean image set $IMG^{LBIG}$ is qualified for being used in training the model $M_k^{DS}$. *defect-sensitive model quality check (DMQC) module*, which is designed for verifying the quality of generated defect-sensitive model $M_k^{DS}$ with the augmented image and label sets $IMG^{AIQC}$ and $L^{AIQC}$. The six modules collaboratively generate a qualified defective bean inspection model with these few amounts of human-labeling images.

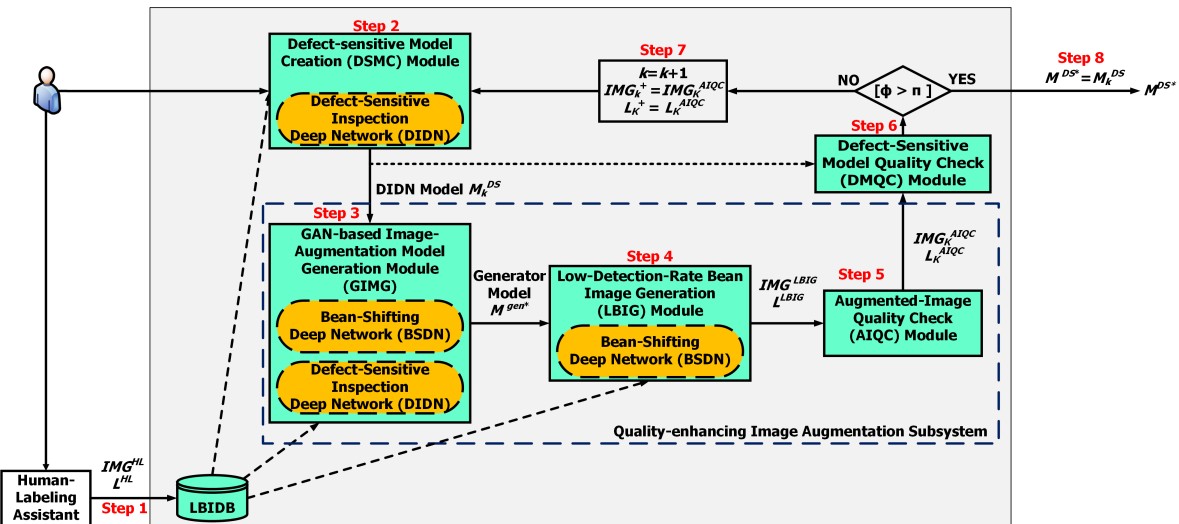

**Figure 4.** The architecture and workflow of the proposed labor-efficient deep learning-based model generation scheme. The output model $M^{DS*}$ plays the role of the defective bean inspection (DBI) model in Figure 3. LBIDB = labeled-bean image database.

The operational flow of these modules is described as below.

**Step 1:** The user labels only a few amounts of sparse defective bean images and stores the labeled bean images in the LBIDB.

**Step 2:** The DSMC module creates a defect-sensitive model for the DIDN based the GAN technique.

**Step 3:** The GIMG module creates a generative network model $M^{gen*}$ for the BSDN and the DIDN.

**Step 4:** The LBIG module generates the augmented labeled defective bean image set (denoted by $IMG^{LBIG}$ and $L^{LBIG}$).

**Step 5:** The AIQC module verifies whether the augmented labeled bean image set $IMG^{LBIG}$ is qualified for being used in training the model $M_k^{DS}$.

**Step 6:** The DMQC module verifies the quality of generated defect-sensitive model $M_k^{DS}$ with the augmented image and label sets $IMG^{AIQC}$ and $L^{AIQC}$.

**Step 7:** Some above augmented bean images that pass the following two quality checks (AIQC and DMQC, presented later) will be used in the next iteration ($k = k + 1$) of training the DSMC module for improving the DIDN model.

**Step 8:** The GALDAM outputs the optimal DIDN model, denoted as $M^{DS*}$, which implements the general DBI model in the DL-DBIS (see Figure 3).

Notice that $k$ used in Step 7 is a global counter used for tracking the training iterations in the framework. Thus, it will frequently appear in the following sections while some historical training data is retrieved. For fast and clearly understand the overall interaction among modules, Table 1 shows how all modules work interactively.

**Table 1.** The interaction relationship of modules in the proposed scheme, as shown in Figure 4.

| Module | Interaction with related modules |
|--------|----------------------------------|
| LBIDB | provides the human-labeling training data to DSMC, GIMG, and low-detection-rate bean image generation (LBIG). |
| DSMC | creates a defect-sensitive model $M_k^{DS}$ for GIMG and DMQC. |
| GIMG | creates a generative network model for LBIG. |
| LBIG | generates $IMG^{LBIG}$ and $L^{LBIG}$ via BSDN for AIQC. |
| AIQC | verifies the quality of $IMG^{LBIG}$ and $L^{LBIG}$ from LBIG and produces $IMG_k^{AIQC}$ and $L_k^{AIQC}$ for DMQC. |
| DMQC | verifies the quality of $M_k^{DS}$ from DSMC with $IMG_k^{AIQC}$ and $L_k^{AIQC}$ and activates DSMC for the next model training iteration if current model is not qualified. |

## 4. Proposed Defect Inspection Models Created from Two Critical Deep Networks and Associated GAN-Structured Data Augmentation with GA-Based Optimizer

Inside the GALDAM, five key mechanisms, including design of Defect-Sensitive Inspection Deep Network (DIDN), design of Bean-Shifting Deep Network (BSDN), GAN-based framework for labeled data augmentation, GA-based optimizer for the proposed GAN framework, and low-detection-rate bean image generation, will be presented in the following subsections. These mechanisms work on two critical deep networks, DIDN and BSDN, and a GAN-structured optimizer, and they collaboratively generate a qualified DIDN model for defect inspection with merely limited human-labeled bean images, by complying the scheme workflow shown in Figure 4.

### 4.1. Design of Defect-Sensitive Inspection Deep Network (DIDN)

During the bean roasting process, some related works [1] show that a defective bean could affect flavor of over 50 normal beans. Thus, our defect-sensitive inspection scheme needs to meticulously differentiate defective beans from normal ones.

Figure 5 shows the architecture of the DIDN, which is an improvement version of our previously designed deep network [36]. The DIDN consists of four main components and each of them are described below. The first component is the dual object feature extractor, used to extract object features from image inputs. For generality of system design, the term "object" is used for representing coffee beans in this paper. This module adopts two methods as underlying feature extraction engines: one is YOLOv3, which is a deep network structure for object feature predictions; the other is Hough circle (HC) transform, which is to extract circular shapes from a given image. Details of the two methods will be presented later. The second component is the HC-based undecidable bounding box pruning module, which is used to pruning those undecidable bounding boxes by using HC information. An undecidable bounding box means more than two or no objects appear stay inside a bounding box, which is generated by the DL model. For many DL-based object detection methods, such situations frequently happen in the intermediate data, and usually the application interface suppresses most these data and uses the case of highest probability for result interpretations. In our scheme, only undecidable bounding boxes are pruning and all decidable bounding boxes are reserved, since they provide information of object confidence and class probabilities for calibrating other overlapped bounding boxes.

The third component is the HC-assisting object identification module, which is used to find out each predicted object with assistance of HCs. Comparing to the YOLOv3, our scheme does not identify an object merely on class probabilities, which may lead to misjudgments if defective beans are heavily overlapped. Our experiments will show the superiority of this module in Section 6. The fourth component is the Gaussian-based calibration module for determining classes of identified objects by calibrating class probabilities of prediction object features. This component determines the bounding boxes of final object estimation. With the proposed DIDN architecture, objects can be accurately detected with two sorts of prediction features and a sequence of post processing on them for improving inspection quality.

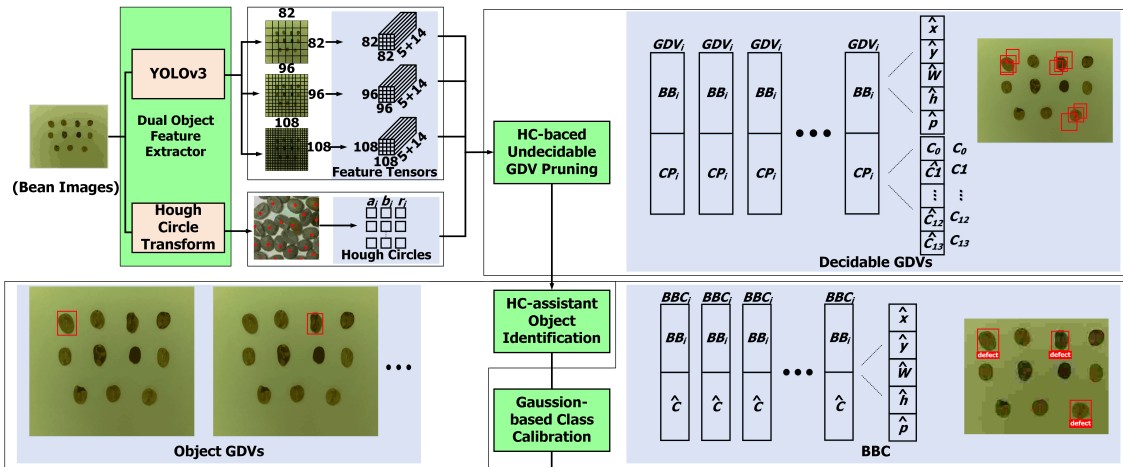

**Figure 5.** The DIDN architecture. Rectangles in light green are functional components, and ones in light blue are the data structures used in the associated components. YOLO = You Only Look Once; HC = Hough circle; GDV = grid-description vector; BBC = bounding box class.

We next briefly present the concepts of YOLOv3 and HC transform, as they are adopted in the dual object feature extractor. The YOLOv3 architecture [37] is adopted in the dual object feature extractor, shown in Figure 5. YOLOv3 totally has 106 convolutional and residual layers, where three scales of prediction features in the tensor forms ($82 \times 82$, $96 \times 96$, $108 \times 108$) are outputted for making final object detection decisions with the non-max suppression (NMS) method [24]. For training a proper YOLOv3 model, the loss function is defined as follows [37]:

$$
\begin{aligned}
Loss =& \lambda_{coord} \sum_{i=0}^{S^2} \sum_{j=0}^{B} \mathbb{1}_{ij}^{obj} \left( (X_{i,j} - \hat{X}_{i,j})^2 + (Y_{i,j} - \hat{Y}_{i,j})^2 \right) \\
&+ \lambda_{coord} \sum_{i=0}^{S^2} \sum_{j=0}^{B} \mathbb{1}_{ij}^{obj} \left( \left( \sqrt{W_{i,j}} - \sqrt{\hat{W}_{i,j}} \right)^2 + \left( \sqrt{h_{i,j}} - \sqrt{\hat{h}_{i,j}} \right)^2 \right) \\
&+ \sum_{i=0}^{S^2} \sum_{j=0}^{B} \mathbb{1}_{ij}^{obj} \left( - \log(\hat{p_{i,j}}) \right) + \lambda_{noobj} \sum_{i=0}^{S^2} \sum_{j=0}^{B} \mathbb{1}_{ij}^{noobj} \left( - \log(1 - \hat{p_{i,j}}) \right) \\
&+ \sum_{i=0}^{S^2} \sum_{j=0}^{B} \mathbb{1}_{ij}^{obj} \sum_{c \in classes} \left( -c \log(\hat{c}) - (1 - c) \log(1 - \hat{c}) \right),
\end{aligned}
\tag{1}
$$

where $\mathbb{1}_{ij}^{obj}$ is an indicator function that determining whether $obj$ is in cell $(i, j)$ and returning 1 if the condition holds; 0 otherwise, $\lambda_{\blacksquare}$ are the pre-defined weighting values for each of the loss terms, $S^2$ are the number of grids of an image used in YOLOv3, and $B$ is the maximum number of bounding boxes in a grid.

The HC transform is a feature extraction technique for detecting circles/eclipses from imperfect digital image inputs [38,39]. Figure 6 shows the idea of the HC transform. In a two-dimensional space, a circle is expressed as follows:

$$
(x - a)^2 + (y - b)^2 = r^2,
\tag{2}
$$

where $(a, b)$ is the center of the circle and $r$ is the radius. If a point $(x, y)$ lies on the circle, then the parameters $(a, b, r)$ can be found according to Equation (2), and the three parameters form a three-dimensional Hough parameter space (see Figure 6b). And all the parameters $(a, b, r)$ that satisfy $(x, y)$ would lie on the surface of an inverted cone whose minimal point is at $(x, y, 0)$. In the 3D Hough parameter space, the circle parameters can be determined by the intersection of multiple conic surfaces

that are generated by all points of the 2D circle. In summary, the HCs provide bean outlines as the second references for precisely identifying beans.

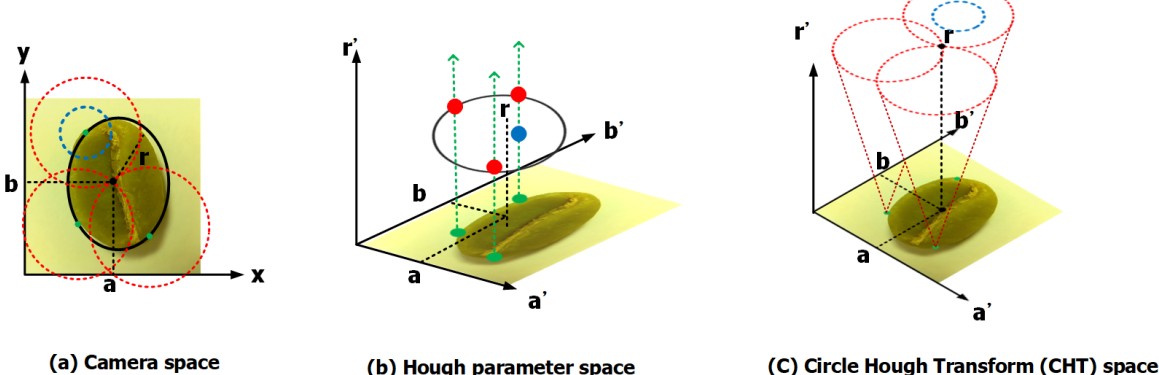

(a) Camera space    (b) Hough parameter space    (C) Circle Hough Transform (CHT) space

**Figure 6.** Illustration of the HC transform in the scenario of detecting a coffee bean.

Two data structures used for data exchange inside the DIDN are presented as follows. A grid-description vector (GDV) shown in Figure 5, indicating prediction features of a grid partitioned by the YOLOv3 to an image, is a primitive data structure used in the components in the DIDN architecture. The GDV format includes two segments. The first segment, called the bounding box (BB) segment, contains five attributes: $\hat{x}, \hat{y}, \hat{w}, \hat{h}, \hat{p}$, where $(\hat{x}, \hat{y})$ indicates the center point, $\hat{w}, \hat{h}$ show the width and the height, respectively, and $\hat{p}$ is the prediction confidence of the associated bounding box. The second segment, call class probability (CP) segment, contains probabilities of all classes. Let $C$ be the number of classes. Then, the second segment contains $C$ attributes, each of which stands for the prediction probability of the corresponding class. For example, 14 classes are used in our previous example in Figure 1, where class 0 stands for the normal beans and other classes 1~13 stand for different kinds of defective beans. For convenient study, the convention $GDV_i(attribute)$ is used for accessing an attribute of $GDV_i$.

The bounding box class (BBC) is a data structure designed for maintaining classes of bounding boxes. The BBC also has two segments. The first segment is the bounding box (BB) segment, which is the same to the first segment of GDVs. The second segment is the class segment, which maintain the object class (i.e., label) of the bounding box. The BBC is the output format of the DIDN.

*4.2. Design of Bean-Shifting Deep Network*

The bean-shifting deep network is a seven-layer neural network for generating new bean labels and their locations by shifting beans in the inputted images. Table 2 shows neural network parameters, mainly including sizes of weighting matrices and bias vectors in all layers, of the BSDN structure in this work (i.e., the generative network shown in Figure 7, which will be discussed later). The BSDN structure includes three transformation layers, a bean-location layer, and three projection layers. Two successive layers in the BSDN are fully connected. The computational effect of processing input data from layer $i - 1$ in a neural layer $i$ can be represented with a linear function followed by an activation function.

Note that the BSDN design is elaborately compromised between the DIDN network and the equipment property. On one hand, for matching the adopted YOLOv3 in DIDN, the BSDN structure considers the number of detection resolution (i.e., $S^2$ in YOLOv3, mentioned in Equation (1)), and the number of detected objects $n$ in a YOLOv3 grid. On the other hand, for matching the camera capacity, the BSDN structure considers the maximum number of beans in a camera-produced image (i.e., $\delta_4$, which will be described in Section 4.5).

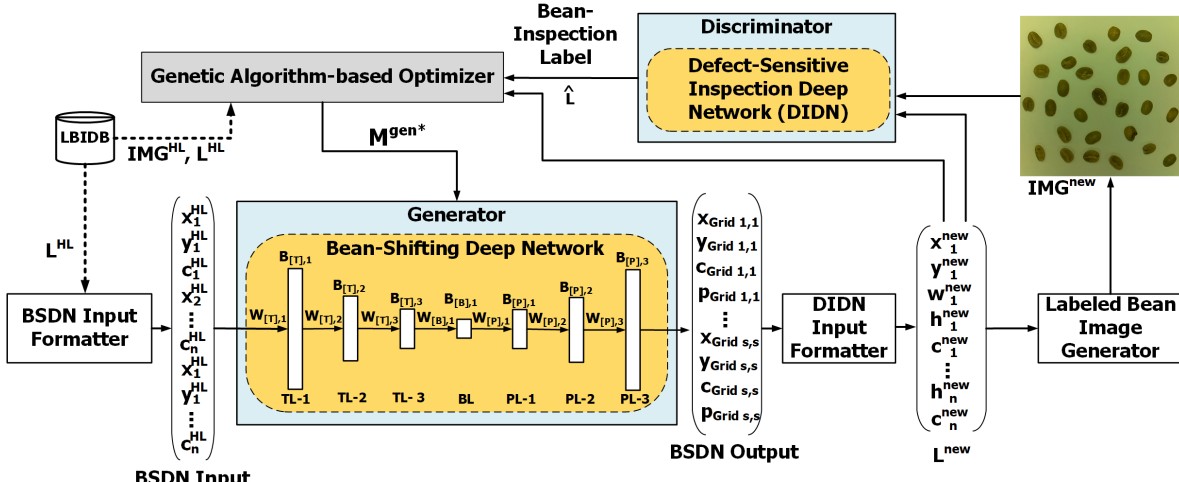

**Figure 7.** The structure and workflow of the generative-adversarial network (GAN)-based data augmentation framework.

**Table 2.** Profile of the bean-shifting deep network structure in this work.

| Layer | # of Neurons | $\lvert W_{\blacksquare}^{\top} \rvert$: Weighting Matrix Size (Weights of Two Consecutive Layers.) | $\lvert b_{\blacksquare} \rvert$: Bias Vector Size (Bias of the Linear Function.) |
|---|---|---|---|
| T-Layer 1 | $2^{\left\lceil log_2^{(\delta_4 \times 3 \times n)} \right\rceil + 2}$ | $2^{\left\lceil log_2^{(\delta_4 \times 3 \times n)} \right\rceil + 2} \times (\delta_4 \times 3 \times n)$ | $2^{\left\lceil log_2^{(\delta_4 \times 3 \times n)} \right\rceil + 2} \times 1$ |
| T-Layer 2 | $2^{\left\lceil log_2^{(\delta_4 \times 3 \times n)} \right\rceil + 1}$ | $2^{\left\lceil log_2^{(\delta_4 \times 3 \times n)} \right\rceil + 1} \times 2^{\left\lceil log_2^{(\delta_4 \times 3 \times n)} \right\rceil + 2}$ | $2^{\left\lceil log_2^{(\delta_4 \times 3 \times n)} \right\rceil + 1} \times 1$ |
| T-Layer 3 | $2^{\left\lceil log_2^{(\delta_4 \times 3 \times n)} \right\rceil}$ | $2^{\left\lceil log_2^{(\delta_4 \times 3 \times n)} \right\rceil} \times 2^{\left\lceil log_2^{(\delta_4 \times 3 \times n)} \right\rceil + 1}$ | $2^{\left\lceil log_2^{(\delta_4 \times 3 \times n)} \right\rceil} \times 1$ |
| Bean-Location Layer | $(\delta_4 \times 3 \times n)$ | $(\delta_4 \times 3 \times n) \times 2^{\left\lceil log_2^{(\delta_4 \times 3 \times n)} \right\rceil}$ | $(\delta_4 \times 3 \times n) \times 1$ |
| P-Layer 1 | $2^{\left\lfloor log_2^{S \times S \times 17} \right\rfloor - 2}$ | $2^{\left\lfloor log_2^{S \times S \times 17} \right\rfloor - 2} \times (\delta_4 \times 3 \times n)$ | $2^{\left\lfloor log_2^{S \times S \times 17} \right\rfloor - 2} \times 1$ |
| P-Layer 2 | $2^{\left\lfloor log_2^{S \times S \times 17} \right\rfloor - 1}$ | $2^{\left\lfloor log_2^{S \times S \times 17} \right\rfloor - 1} \times 2^{\left\lfloor log_2^{S \times S \times 17} \right\rfloor - 2}$ | $2^{\left\lfloor log_2^{S \times S \times 17} \right\rfloor - 1} \times 1$ |
| P-Layer 3 | $(S \times S \times 17)$ | $(S \times S \times 17) \times 2^{\left\lfloor log_2^{s \times s \times 17} \right\rfloor}$ | $(S \times S \times 17) \times 1$ |

Table 3 shows linear and activation functions of the BSDN, which determine decision behavior of neurons and layers in the deep network. A linear function for each neuron is computed with a weighting matrix $\mathbf{W}_{[z],i}^{\top}$ and a bias vector $\mathbf{b}_{[z],i}$ by the following equation.

$$\mathbf{u}_i = \mathbf{W}_{[z],i}^{\top} \mathbf{q}_{i-1} + \mathbf{b}_{[z],i} \tag{3}$$

$$\text{where } z = \begin{cases} T, & \text{if } i = 1, 2, 3, \ (T \text{ stands for transformation layers}) \\ B, & \text{if } i = 4, \ (B \text{ stands for the bean-location layer}) \\ P, & \text{if } i = 5, 6, 7, \ (P \text{ stands for projection layers}) \end{cases}.$$

Note that $\mathbf{W}^{\top}$ and $\mathbf{b}$ follow conventions in popular DL papers. The activation function of a neuron is performed with the sigmoid function, represented as $\mathbf{q}_i = \sigma(\mathbf{u}_i)$, and the $\sigma(\cdot)$ is defined as follows:

$$\sigma(x) = \frac{1}{1 + e^{-x}}, \tag{4}$$

where $x$ stands for any possible input $\mathbf{u}_i$ and $e$ denotes Euler's number.

After processing of the BSDN, the output of the final layer is interpreted as the new generated labels and locations for further use.

**Table 3.** Sketch of linear and activation functions for the BSDN in this work. The input $\mathbf{q}_0 = [x_i^{HL}, y_i^{HL}, c_i^{HL}, 0, \cdots, 0]^\top, i = 1, \dots$ with size $(\delta_4 \cdot 3 \cdot n) \times 1$. Fill 0 to the tail of $\mathbf{q}_0$ if input beans are insufficient.

| Layer | Linear Function | Activation Function $\sigma(\bullet)$: See Equation (4). |
|---|---|---|
| T-Layer 1 | $\mathbf{u}_1 = \mathbf{W}_{[T],1}^\top \mathbf{q}_0 + \mathbf{b}_{[T],1}$ | $\mathbf{q}_1 = \sigma(\mathbf{u}_1)$ |
| T-Layer 2 | $\mathbf{u}_2 = \mathbf{W}_{[T],2}^\top \mathbf{q}_1 + \mathbf{b}_{[T],2}$ | $\mathbf{q}_2 = \sigma(\mathbf{u}_2)$ |
| T-Layer 3 | $\mathbf{u}_3 = \mathbf{W}_{[T],3}^\top \mathbf{q}_2 + \mathbf{b}_{[T],3}$ | $\mathbf{q}_3 = \sigma(\mathbf{u}_3)$ |
| Bean-Location Layer | $\mathbf{u}_4 = \mathbf{W}_{[B],1}^\top \mathbf{q}_3 + \mathbf{b}_{[B],1}$ | $\mathbf{q}_4 = \sigma(\mathbf{u}_4)$ |
| P-Layer 1 | $\mathbf{u}_5 = \mathbf{W}_{[P],1}^\top \mathbf{q}_4 + \mathbf{b}_{[P],1}$ | $\mathbf{q}_5 = \sigma(\mathbf{u}_5)$ |
| P-Layer 2 | $\mathbf{u}_6 = \mathbf{W}_{[P],2}^\top \mathbf{q}_5 + \mathbf{b}_{[P],2}$ | $\mathbf{q}_6 = \sigma(\mathbf{u}_6)$ |
| P-Layer 3 | $\mathbf{u}_7 = \mathbf{W}_{[P],3}^\top \mathbf{q}_6 + \mathbf{b}_{[P],3}$ | $\mathbf{q}_7 = \sigma(\mathbf{u}_7)$ |

### 4.3. GAN-Based Framework for Labeled Data Augmentation for the GIMG Module

Since the number of human-labeled images is limited for minimizing human effort, we proposed a GAN-based data augmentation mechanism to generate certain new images for training the optimal DIDN model. Notice that the generative adversarial network used in this work contains two subnetworks of inconsistent data formats, which incurs the difficulty in the training process for optimizing parameters of the GAN parameters. Hence, we further proposed a genetic algorithm (GA)-based optimizer for training such GAN of inconsistent data formats.

Figure 7 illustrates the workflow of the GAN-based data augmentation mechanism, which consists of the BSDN input formatter, two deep networks (BSDN and DIDN), the labeled bean image generator, the DIDN input formatter, and a GA-based optimizer. The BSDN input formatter converts $L^{HL}$ to the condensed bean label format, which contains center locations and class attributes of beans, i.e., $(x_i, y_i, c_i)^T, i = 1, \cdots, n$. The discriminator network is designed to be the same to the DIDN with the model $M^{DS}$ obtained in previous section. The generator network is designed to shift beans of the input image to different locations for automatically augmenting new labeled bean images. The specification of the generator network will be presented later. The labeled bean image generator is to render new images of beans according to the new locations and classes indicates in the output of the generative network, together with bean labels for the new rendered images. The DIDN input formatter is to transform the BSDN output in the condensed bean label form to the DIDN input format in the complete bean label form, which also can be used to render a corresponding bean image, denoted by $IMG^{new}$, shown in the figure. The GA-based optimizer that contains several GA-related components shown in the figure is designed to find out the optimal generator network model $M^{gen*}$ (i.e., the BSDN in this work) with the bean-inspection label $\hat{L}$ from the DIDN, $L^{new}$, $IMG^{new}$, $IMG^{HL}$, and $L^{HL}$, on which technical details will be presented later.

### 4.4. GA-based Optimizer for the Proposed GAN Framework

Notice that the data formats between the generative network and the discriminator network in our GAN framework are different in this work, where the generative network outputs bean labels and the discriminator network uses inputs of bean images. This inconsistent data formats incur stochastic gradient decent (SGD) techniques cannot directly applied to the model optimization process in our GAN framework, because the derivative between the two networks cannot estimated during finding optimal neural network parameters. To alleviate this issue, we proposed the GA-based optimizer used for finding optimal model for our GAN framework, whose component structure is illustrated in Figure 8. The design of the GA-based optimizer is described as follows.

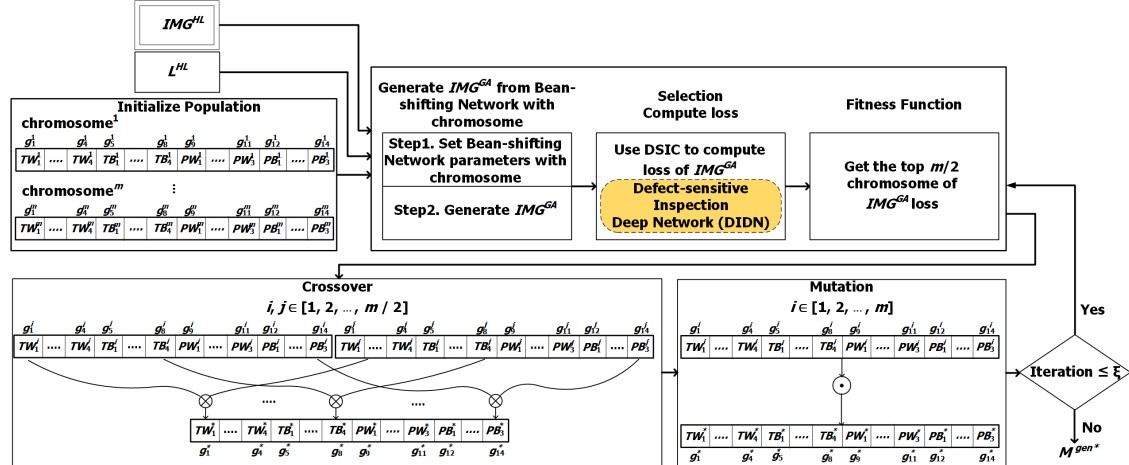

**Figure 8.** The structure and workflow of the genetic algorithm (GA)-based optimizer. This figure describes the corresponding module design in the top-left corner of Figure 7.

**Chromosome (denoted $\mathcal{C}^i$)**: represents a phenotype of an individual, and it consists of 14 genes corresponding to a BSDN structure for representing parameters of a BSDN, i.e., $\mathcal{C}^i = (g_1^i, \cdots, g_{14}^i)$, and each gene $g_{i,j}$ is defined as follows:

$$
g_j^i = \begin{cases}
\mathbf{W}_{[T],j}, & \text{if } j = 1, 2, 3, \\
\mathbf{b}_{[T],j}, & \text{if } j = 4, 5, 6, \\
\mathbf{W}_{[B],j}, & \text{if } j = 7, \\
\mathbf{b}_{[B],j}, & \text{if } j = 8, \\
\mathbf{W}_{[P],j}, & \text{if } j = 9, 10, 11, \\
\mathbf{b}_{[P],j}, & \text{if } j = 12, 13, 14.
\end{cases}
\tag{5}
$$

**Crossover**: generates new offspring with two randomly selected chromosomes (say, $\mathcal{C}^i$ and $\mathcal{C}^j$). Since sizes of genes are not equal, the traditional single-point crossover needs to be modified to be adaptive to such situation and still preserve the semantics of genes. Let $\mathcal{C}^* = \{g_k^* | k = 1, \ldots, 14\}$ be the chromosome of the new offspring. The gene $g_k^*$ comes from the gene-wise single-point crossover over matrix (SPCOM) of the selected parent chromosomes, i.e., $g_k^* = g_k^i \otimes g_k^j$, where $\otimes$ is the SPCOM operator. Figure 9 illustrates the key processing steps of the SPCOM. Elements of the two-dimensional matrix representation of the BSDN parameters are re-arranged into the one-dimensional chromosome representation, and then the single-point crossover can be performed over genes. After the crossover, the new generated chromosomes can be transformed back to the two-dimensional matrix representation of the BSDN parameters.

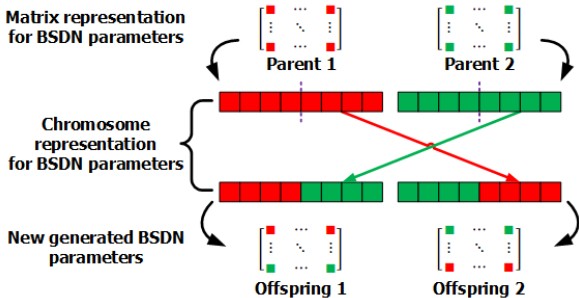

**Figure 9.** The gene-wise single-point crossover over matrix of the selected parent chromosomes.

**Mutation**: slightly modifies selected chromosomes for avoiding local minimum. Genes $g_j^i$ of the selected chromosome $\mathcal{C}^i$ are mutated with the bit string mutation method. The mutation of $\mathcal{C}^i$ is denoted as $\mathcal{C}^i = \odot \mathcal{C}^i$ with the mutation operator $\odot$ as shown in Figure 8.

**Fitness function**: calculates the fitness score for each chromosome as follows. Genes $(g_1^i, \cdots, g_{14}^i)$ of a chromosome $\mathcal{C}^i$ are converted to BSTN structure parameters, where the corresponding BSTN can then generate a labeled data set $(IMG_i^{GA}, L_i^{GA})$ with the bean image render module. Next, the current defect inspection deep network with the model $M_k^{DS}$ are used to calculate the loss values of the data sets $IMG_i^{GA}$ and $L_i^{GA}$, where the loss values can be obtained with the DIDN. The mean of these loss values from the data sets $IMG_i^{GA}$ and $L_i^{GA}$ is the fitness value of the chromosome.

**Selection method**: determines which chromosomes can survive in the population of the GA procedure. Once the fitness function is defined, all chromosomes can be sorted in ascending order according to their fitness values. The last half of chromosomes are discarded in each iteration of the GA procedure.

These are the main designs of the GA used in this optimizer. Algorithm 1 summarizes the detailed steps for the GA-based GAN optimizer. The GA process terminates after pre-defined iterations $\xi$. After the GA process, the optimal chromosome found can be converted into the corresponding BSDN model $M^{gen*}$ as the optimizer output.

---

**Algorithm 1:** GA-based GAN Optimizer.

**Input:** The models of Bean-Shifting Deep Network;

**Output:** The optimal BSDN model $M^{gen*}$;

```
// Initialization
```
Initialize population (create random chromosomes): $G(0)$ and set $i = 0$; /* i:  iteration
    counter for GA optimizer.                                             */

**while** *($i \leq \xi$)* **do**
```
    /* termination condition is not met yet.                            */
```
    Map each individual (BSDN model) in the population to its phenotype (matrix representation);
    Evaluate fitness score for each individual, where the evaluation involves the DIDN to compute loss function for an individual;
    Select parents from $G(i)$ based on their fitness in the population;
    Apply crossover/mutation to parents and generate offsprings; `// ref.  to Figure 9.`
    Select individuals from $G(i)$ and offsprings based on their fitness, and add the selected ones into $G(i+1)$;
    $i = i + 1$;
**end**

Set $\mathcal{C}^*$ is the chromosome with maximum fitness in the current population;
Transform $\mathcal{C}^*$ to the BSDN model $M^{gen*}$; `// ref.  to Figure 9.`
**return** $M^{gen*}$;

---

### 4.5. Low-Detection-Rate Bean Image Generation for the LBIG Module

The low-detection-rate bean image generation mechanism placed in the LBIG module will generate bean images by using the BSDN with the generative model. This mechanism generates bean images with low detection rate, so that these specially rendered bean images whose locations are determined by the BSDN can assist the DMSC module to continuously improve the DIDN model $M_k^{DS}$.

The low-detection-rate bean image generation mechanism generates four new augmented image sets of different bean density types ($SD, LD, MD, HD$ mean same/low/medium/high densities, respectively) and proceeds as follows. For convenient study, $\delta_i$ is denoted as the number of beans in a generated image for the density type $i$, where the value of subscript $i$ stands for the four types ($SD$, $LD$, $MD$, $HD$), respectively). In this work, $\delta_i$ is set to $1/2/4/6$-times average number of beans in human-labeled images. This mechanism generates augmented image set for four density types

$(SD, LD, MD, HD)$ in sequence. Firstly, $\delta_i$ beans and labels are randomly selected from human-labeling data, i.e., $IMG^{HL}$ and $L^{HL}$. Secondly, we organize the selected beans and labels as a BSDN input and use the BSDN with the model $M_k^{DS}$ to generate new locations of these beans. Notice that the BSDN trained in the GAN-based framework shifts the beans in the input to locations that are less successful detection rate to the DIDN with the model $M_k^{DS}$. Thus, the DIDN can improve its inspection ability with such generated images. Thirdly, we render a new image according to the new locations of beans. These steps are repeated until $\delta_i$ images of each density type are generated.

## 5. Proposed Model Quality Control in the GALDAM

### 5.1. Augmented-Image Quality Checking for Filtering Heavily Bean-Overlapping Images

The augmented-image quality checking mechanism is designed to verify whether the augmented images can be the qualified in training models $M_k^{DS}$ for the defect-sensitive inspection network. The criteria is that the overall overlapping area of beans in a single bean image shall not be greater than a threshold value, represented by $\theta^{AIQC}$, since the too high degree of bean overlapping decreases inspection accuracy of the trained model. More precisely, the criteria of reserving an augmented image can be expressed as the following equation:

$$\sum_{m=1}^{\gamma_i-1} \sum_{n=m+1}^{\gamma_i} IoU(b_m^{LBIG}, b_n^{LBIG}) \begin{array}{l} \leq \theta^{AIQC}, \quad \text{reserve the augmented image.} \\ > \theta^{AIQC}, \quad \text{discard the image.} \end{array} \tag{6}$$

where the $\gamma_i$ is defined as number of GDVs in image $i$ and $b_j^{LBIG}$ is denoted $j$-th bean image in LBIG. The IoU (Intersection over Union) [24] means the degree of two overlapped areas $Area_i$ and $Area_j$ is measured, and defined as

$$IoU(Area_i, Area_j) = \frac{|Area_i \cap Area_j|}{|Area_i \cup Area_j|}, \tag{7}$$

where $|Area_x|$ indicates the area size of $Area_x$. Based on this idea, we developed a procedure for checking quality of all augmented bean images.

The detailed steps of the augmented-image quality checking procedure are described as follows.

**Step 1:** Set image counter $i = 0$, bean label set $L^{AIQC} = \varnothing$, and bean image set $IMG^{AIQC} = \varnothing$.
**Step 2:** Determine whether image $i$ is qualified to be reserved. If the criterion ($\sum_{e=1}^{\gamma_i-1} \sum_{f=e+1}^{\gamma_i} IoU(b_e^{LBIG}, b_f^{LBIG}) < \theta^{AIQC}$) holds, then the $i$-th bean image is qualified and reserves its information as the following substeps:
　　Add $L_i^{LBIG}$ into $L^{AIQC}$;
　　Add $IMG_i^{LBIG}$ into $IMG^{AIQC}$;
**Step 3:** Prepare the next image. If $i$ is less than the number of training images, then increase the image counter by one, i.e., $i = i + 1$ and go to Step 2.
**Step 4:** Return the qualified images and associated labels, $(IMG^{AIQC}, L^{AIQC})$.

Figure 10 shows examples of bean-overlapping images that do not pass the augmented-image quality checking criterion. By looking the figure, we can see that those heavily overlapping of beans are not even easily distinguished by human experts, and thus, it is reasonable to discard such images. The formula above, i.e., Equation (6) is designed to appropriately achieve this goal by using the parameter $\theta_{AIQC}$, which determine the acceptable overlapping degree.

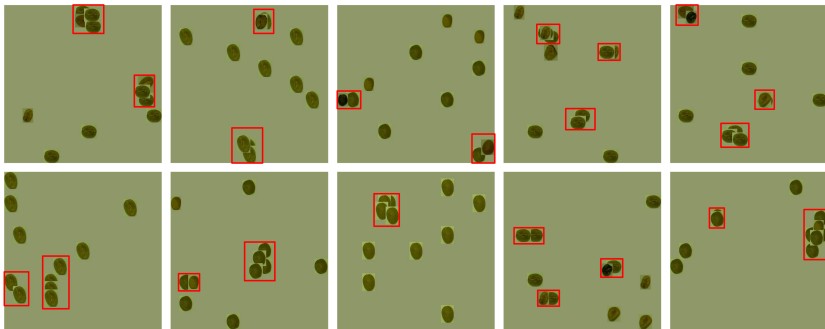

**Figure 10.** Some instances of heavily bean-overlapping images that are discarded. The overlapping area of beans is marked in red rectangles.

### 5.2. Model Quality Checking for Continuously Improving Inspection Capability

The model quality checking mechanism is designed to determine whether the defect inspection network model $M_k^{DS}$ can be further improve or not. If $M_k^{DS}$ performs stable in accuracy, then the $M_k^{DS}$ is the optimal model, used for defective bean inspection in the robotic arm system. The criterion for implementing this idea is modeled as that difference values of the loss function of the defect inspection network with model $M_k^{DS}$ by feeding $(IMG^{AIQC}, L^{AIQC})$ are smaller than a threshold $\theta^{DMQC}$ after consecutive $\pi$-times checks. More precisely, the criterion of evaluating whether the model training is stopped or not can be expressed as the following equation.

$$\sum_{i=k-(\pi-1)}^{k} [\![\phi_i < \theta^{DMQC}]\!] - \pi \begin{array}{l} \geq 0, \quad \text{Current model is qualified. Quit the training process.} \\ < 0, \quad \text{Continue the model training process,} \end{array} \tag{8}$$

$$\text{where } \phi_k = \frac{loss(IMG_k^{AIQC}|M_k^{DS}) - loss(IMG_{k-1}^{AIQC}|M_k^{DS})}{loss(IMG_{k-1}^{AIQC}|M_k^{DS})}.$$

Note that $[\![x]\!]$ is a Boolean operator which equals 1 if condition $x$ holds, otherwise 0, and $loss(D|M)$, stands for the loss function of the defect-inspection network with the model $M$ by feeding the data set $D$.

The detailed steps of the model quality checking procedure are described as follows.

**Step 1:** Calculate $\phi_k$ in Equation (8): $\phi_k = \frac{loss(IMG_k^{AIQC}|M_k^{DS}) - loss(IMG_{k-1}^{AIQC}|M_k^{DS})}{loss(IMG_{k-1}^{AIQC}|M_k^{DS})}$.

**Step 2:** Check model quality by computing the criterion: $\sum_{i=k-(\pi-1)}^{k} [\![\phi_i < \theta^{DMQC}]\!] \stackrel{?}{=} \pi$.

    **Step 2.1:** If the condition $\sum_{i=k-(\pi-1)}^{k} [\![\phi_i < \theta^{DMQC}]\!] = \pi$ holds, then the current model is the best model, which is the output of this algorithm. That is, two following steps is performed in this subcase: $M^{DS*} = M_k^{DS}$;

    Return $M^{DS*}$;

    **Step 2.2:** Otherwise, the whole model training process shall continue with preparing new data set for the following iteration: $k = k + 1$;

    $IMG_k^+ = IMG^{AIQC}$;

    $L_k^+ = L^{AIQC}$;

    go back to the DSMC module; (see Figure 4).

Notice that the parameter $\pi$ controls the degree of model quality, and the administrator can determine $\pi$ according to application requirements. The large $\pi$ would find out if a model is performing more stably, but it spends more training time, and vice versa.

## 6. Case Study

### *6.1. Experimental Settings and Performance Metrics*

We deployed the proposed labor-efficient GAN-based model generation scheme, including all five main modules presented in Section 3, on both the edge-device-controlled robotic system and the GPU-based server for training deep learning networks used in our scheme. On the robotic system, a Raspberry Pi equipped with an Intel Movidius Neural Compute Stick and an 800M-dpi camera (OV5647) inspects beans with the DIDN plus the founded optimal model $M^{DS*}$ and guides the robotic arm to pick off detected defects [20]. The end effector of the arm is an air valve with a 90 Kpa pump. Our proposed scheme is implemented with mixture of Tensorflow, Python, and related tools, for generating the optimal DIDN model $M^{DS*}$ on a desktop computer with Intel Core i5-8500 3.0 GHz, 24 GB RAM, and a NVIDIA GTX 2080 Ti GPU card. For comparison, we also implemented two defect inspection schemes with the HCADIS [36] and the YOLOv3 [24,37].

Figure 11 shows experimental equipment, which is a robotic arm with a camera on the end effector for removing defective coffee beans used in this case study (Authors from seven Taiwan universities share system development load and accomplish experiments in the case study). The arm device is uArm Swift Pro 3-axes robotic arm. The camera is the Raspberry Pi camera OV5647 with maximal 1080p resolution. The arm controller device is a single-chip computer of Raspberry Pi 3 with 1.2 GHz processor, 1 GB RAM, and 16 GB disk. The software modules mentioned in Section 3 are developed with Python, and some image processing functions are achieved with the OpenCV library [40]. The size of the inspection area is $17.5 \times 12.5$ cm$^2$, which is limited by the camera capacity. The height of the robotic operational space is 25 cm, which can contain around 100 beans in the inspection area. Certain data sets captured by the camera OV5647 for training and testing in YOLOv3 are shown in Figure 12. Defective and normal beans are randomly placed. In this case study, 30 bean images are used for training, and 10 bean images for testing. Beans are labeled by experts with the 14 classes shown in Figure 1.

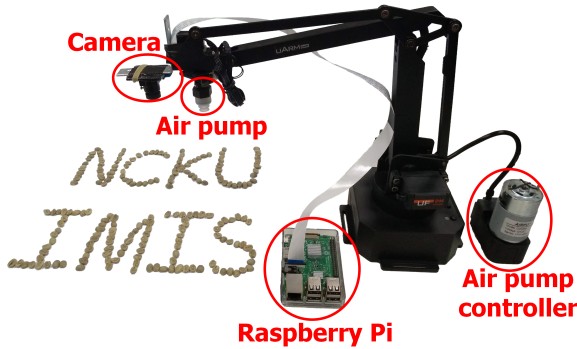

**Figure 11.** Illustration of experimental equipment. A robotic arm with a camera on the end effector for removing defective coffee beans is used in this work.

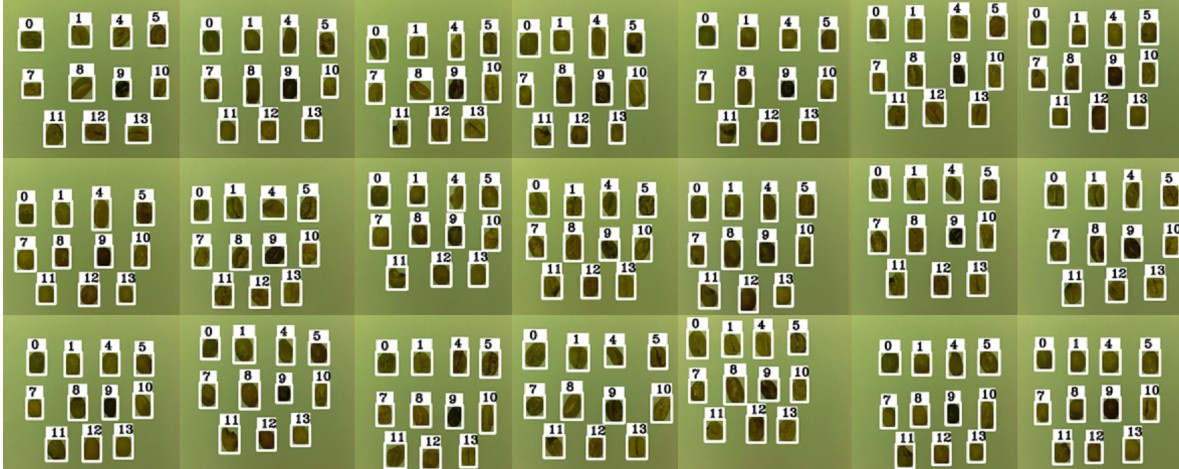

**Figure 12.** Illustration of some training data set. The white boxes in each image stands for defective beans marked by bean-picking experts.

For evaluating the effectiveness of different methods, three performance metrics are used in this paper. The first metric is Defect Inspection Precision (DIP), which is to measure the number of inspected defective beans under the used bounding boxes. The DIP equals the percentage of the true-positive predictions over all positive ones weighted by the percentage of true-positive bounding boxes over all bounding boxes and is expressed as the following formula:

$$
\text{DIP} = \left( \frac{\sum \llbracket BBT_i(\hat{c}) \neq 0 \& label(BBT_i) \neq 0 \rrbracket}{\sum \llbracket BBT_i(\hat{c}) \neq 0 \& label(BBT_i) \neq 0 \rrbracket + \sum \llbracket BBT_i(\hat{c}) \neq 0 \& label(BBT_i) = 0 \rrbracket} \right) \times \left( \frac{\sum \llbracket BBT_i(\hat{c}) \neq 0 \& label(BBT_i) \neq 0 \rrbracket}{\sum \llbracket BBT_i(\hat{c}) \neq 0 \rrbracket} \right) , \quad (9)
$$

where $BBT_i(\hat{c})$ denotes result of defect inspection algorithm, $label(BBT_i)$ denotes ground truth of bean, and $\llbracket x \rrbracket$ is a Boolean operator which equals 1 if condition $x$ holds, otherwise 0.

The second metric is Defect Inspection Recall (DIR), which is to measure the number of defective beans that are not inspected yet under the used bounding boxes. The DIR is equal to the percentage of the true-positive predictions over all defective beans weighted by the percentage of true-positive bounding boxes over all bounding boxes and is expressed as the following formula:

$$
\text{DIR} = \left( \frac{\sum \llbracket BBT_i(\hat{c}) \neq 0 \& label(BBT_i) \neq 0 \rrbracket}{\sum \llbracket BBT_i(\hat{c}) \neq 0 \& label(BBT_i) \neq 0 \rrbracket + \sum \llbracket BBT_i(\hat{c}) = 0 \& label(BBT_i) \neq 0 \rrbracket} \right) \times \left( \frac{\sum \llbracket BBT_i(\hat{c}) \neq 0 \& label(BBT_i) \neq 0 \rrbracket}{\sum \llbracket BBT_i(\hat{c}) \neq 0 \rrbracket} \right) , \quad (10)
$$

where $BBT_i(\hat{c})$ denotes result of defect inspection algorithm and $label(BBT_i)$ denotes ground truth of bean.

The third metric is Defect Inspection Accuracy (DIA), which is to measure the inspected area of defective beans over all inspection area. The DIA is equal to the degree of overlapped area between a defective bean and inspection area for all true-positive predictions over the number of true-positive ones and is expressed as the following formula:

$$
\text{DIA} = \frac{\sum \left( \llbracket BBT_i(\hat{c}) \neq 0 \& label(BBT_i) \neq 0 \rrbracket \times IoU(Area(BBT_i), Area(obj_i)) \right)}{\sum \llbracket BBT_i(\hat{c}) \neq 0 \& label(BBT_i) \neq 0 \rrbracket}, \quad (11)
$$

where $BBT_i(\hat{c})$ denotes result of defect inspection algorithm and $label(BBT_i)$ denotes ground truth of bean.

*6.2. Visualization of Data-Augmentation Results*

Figure 13a–d shows the augmented bean images of different densities for being used in the training stage. Figure 13a is the original image labeled by a human, and Figure 13b–d are generated with different densities by our proposed LBIG module presented in Section 4.5. The generated images contain different densities, which are categorized into low, medium, and high-density coffee beans, respectively, in our work. We visualize some of them in the figure for illustrating what is happened in the stage of executing the LBIG module. From the figure, it is clearly that our scheme indeed generate the very different augmented bean images to be used in the training stage for increasing defect inspection capacity of the DIDN model. We design experiments presented later to study whether these generated images assist the DIDN to train models of superior identification performance.

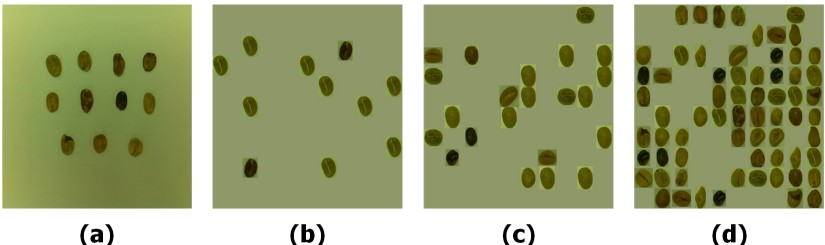

**(a)**          **(b)**          **(c)**          **(d)**

**Figure 13.** Augmented bean images: (**a**) original, (**b**) low density, (**c**) medium density, (**d**) high-density.

Figure 14 shows visualization results of inspection effects for the YOLOv3, HCADIS, and our proposed DL-DBIS. The white bounding boxes are ground truths of defective beans, while the red ones are results of the respective defect inspection algorithm. Due to the length limit, only four instances are shown here. By looking these visualized results, our proposed DL-DBIS obviously discovers most defective beans, comparing to YOLOv3 and HCADIS. That is, DL-DBIS indeed significantly improves quality of defective bean inspection in YOLOv3 and HCADIS. The detailed experimental statistics will be presented later.

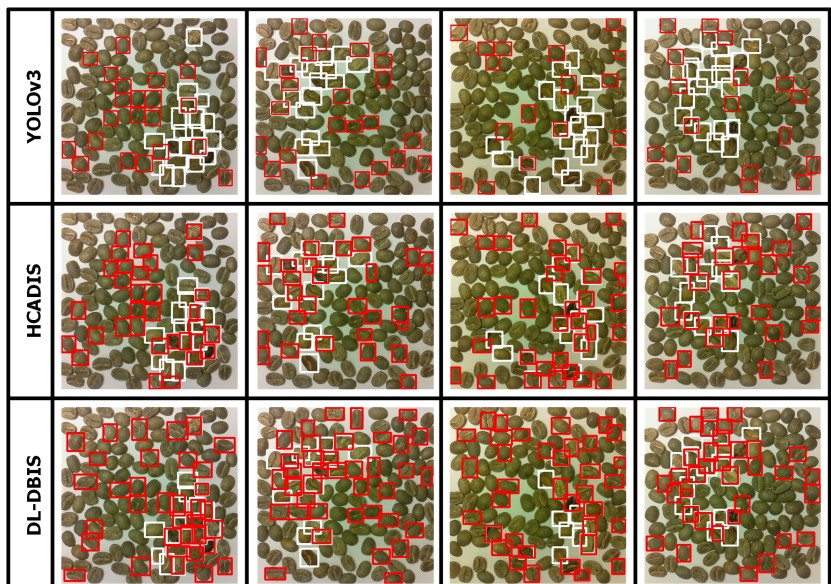

**Figure 14.** Visualized effects of different algorithms. Four bean images are used to illustrate the results of YOLOv3 [24,37], HCADIS [36], and our proposed DL-DBIS.

*6.3. Performance Study of the Optimal Model in the DIDN*

This experiment is designed to verify the effectiveness of the DBI model shown in Figure 3, where the implementation instance in the GALDAM is the optimal model of the DIDN (i.e., $M^{gen*}$)

discussed in Section 4.4. Figure 15 shows the performance study of the optimal model $M^{gen*}$ of the DIDN to ten randomly selected bean images in term of DIP, DIR, and DIA, respectively.

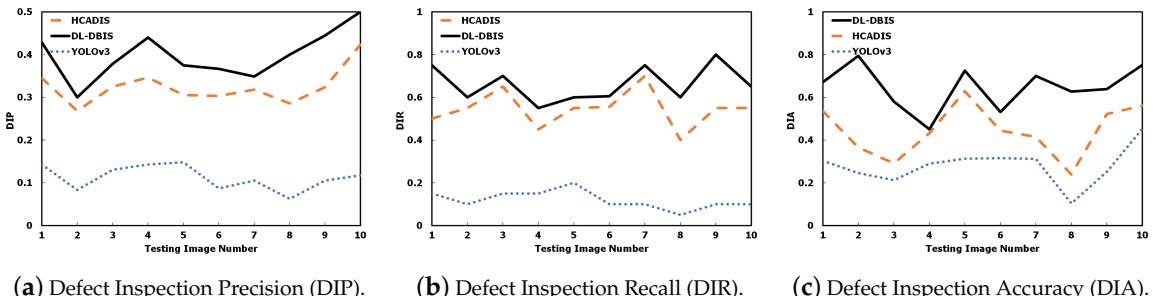

(**a**) Defect Inspection Precision (DIP).  (**b**) Defect Inspection Recall (DIR).  (**c**) Defect Inspection Accuracy (DIA).

**Figure 15.** Performance study of the DIDN to ten randomly selected bean images.

The DIP values stand for the ratio that classifying normal beans into defects, and the higher DIP values signify that a defect inspection scheme makes less mistakes. From the experimental results shown in Figure 15a, DL-DBIS has higher DIP values than HCADIS and YOLOv3, meaning that our proposed scheme is the best among the three. HCANDIS uses the HC transform to calibrate the prediction results from YOLOv3. Since most beans are of the circular shape, HCANDIS can successfully calibrate many mistakes of YOLOv3 and thus performs better than YOLOv3. Our proposed DL-DBIS can automatically generate new bean images in the training stage through the LBIG module and further prune the unnecessary bounding boxes by using HC transformation. Since low quality images are pruned by the model quality control in DL-DBIS, those augmented bean images can effectively increase the precision of the bean inspection model. Therefore, the DL-DBIS performs superior than HCADIS and YOLOv3.

The DIR values stand for the ratio that the number of predictive defective beans to the number of actual defective ones, and the higher DIR values signify that a defect inspection scheme lose less defective beans. Comparing to the DIP, the DIR does not consider the cases that classifying normal beans into defects, which makes the DIR be the index to verify the bean quality after bean picking process by a specified scheme. From the experimental results shown in Figure 15b, DL-DBIS has much higher DIP values than HCADIS and YOLOv3, and most cases are over 60%, even approaching 80%. The results show that our proposed scheme has the best capacity to discover defective beans. This also means that if a client requests to remove all defective beans, our proposed scheme will be the best choice among the three to achieve this task. The key successful factor is that the DL-DBIS can generate effective new bean images, where other two cannot achieve. The high-quality augmented bean images not only shorten the training time, but also avoids the unconvergence problem in training the deep network, DIDN.

The DIA values stand for the ratio that the area of predictive defective beans to the area of actual defective ones. That is, the DIA measures the overlapping degree of correctly predictive defective beans. The higher DIA values signify that a defect inspection scheme has higher probability to successfully remove defective beans. From the experimental results shown in Figure 15c, DL-DBIS has much higher DIA values than HCADIS and YOLOv3, and most cases are by 80%, whereas YOLOv3 only by 20%. Similar to previous studies, the success of the DL-DBIS comes from generating high-quality new bean images. The high DIA implies that the arm head has less probability of sideswiping beans nearby, which might muddle next rounds of bean picking.

*6.4. Efficiency of the GA-Based Optimizer*

This experiment studied the ability of augmenting new bean images in the quality-enhancing image augmentation subsystem, where the key of determining content of new images is the GA optimizer in the GIMG module (see Figure 4). Figure 16 shows the fitness value as a function of the

GA iterations, where the fitness values are the average loss values of DIDNs in which models come from transforming each chromosome.

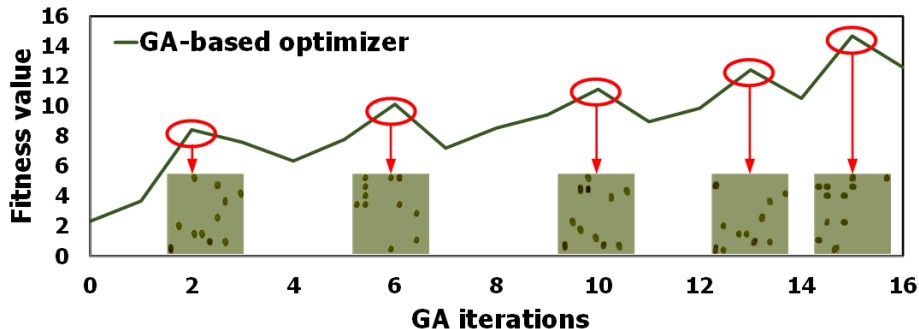

**Figure 16.** The fitness value as a function of the GA iterations.

The experimental result shows that the trend of fitness values grows with the GA iterations, meaning the GA optimizer indeed finds out new chromosomes that corresponding BSDNs can generate bean locations with low detection rate in the DIDN. This meets the design purpose of the GA optimizer. Some bean image instances from different GA iterations also shown in the figure, and these instances indeed look very different, which provides the visual evidences of high-quality augmented bean images used for training better models. Some more clear augmented bean images can be seen in Figure 13.

The GA optimizer consumes the resources of CPU and RAM. The time cost of the GA optimizer is not considered in the experiments as it is not the performance bottleneck of the DL-DBIS. On the other hand, the model training for the DIDN and BSDN spends most time, which is the actual performance bottleneck of the DL-DBIS, and we will discuss its time cost later.

### 6.5. Performance Comparisons of Various Schemes to Different Number of Human-Labeled Images

Figure 17 shows the performance comparisons of various defect inspection schemes to different number of human-labeled images. The horizontal axis is the number of inputted images, which varies from 5 to 20 human-labeled bean images, and the vertical axes are the DIP, DIR, and DIA defined above in subfigures, respectively. Due to the data augmented method GALDAM used in the proposed DL-DBIS, the numbers of generated bean images used in the training are also placed in the parentheses, i.e., 23, 46, 71, 97 in each case, respectively, for further reference. Recall that there is model quality control in DL-DBIS, and these automatically generated images have properties of low degree of bean overlapping and low detection rate. This is the reason why these numbers are not rounded integers.

From the results, all schemes have high DIP, DIR, and DIA values as the increase of inputted bean images, which are consistent to our intuitive, which shows the effectiveness of the three metrics. The HCADIS perform better than YOLOv3 as it considered the roundness property of coffee beans inside its mechanism. Among the three schemes, the proposed DL-DBIS performs best, which show the proposed DL-DBIS further improve the defect identification capacity by augmenting bean images. Furthermore, from our tests over this data set, the augmented images assist the defect identification model to find out almost all defective beans as the DIR is over 0.8 from Figure 17b. The DL-DBIS also accurately identify defective beans as its DIA is very close to 0.7 from Figure 17c. Our labeled beans are too similar and are of too low quantity, the precision (DIP) is low meaning that the model may make mistakes for unseen beans, but DL-DBIS still performs better than other two schemes under the same inputted data size.

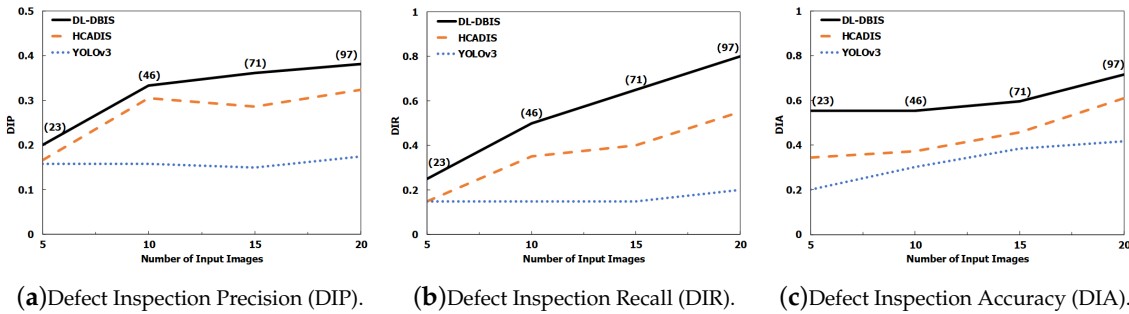

(**a**)Defect Inspection Precision (DIP).　　(**b**)Defect Inspection Recall (DIR).　　(**c**)Defect Inspection Accuracy (DIA).

**Figure 17.** Performance comparisons of different defect inspection schemes to various number of inputted bean images.

### 6.6. Efficiency of the Proposed DL-DBIS

The second experiment studies the labeling time between DL-DBIS and human labeling from Figure 18. On human labeling, an expert of defective bean removal is trained to label around 35 bean images, and the average time of labeling an image is used to estimate the required time from 100 to 800 images. Thus, the estimated human-labeling time is a straight regression line. On the DL-DBIS, since two networks (the DIDN and the BSDN) need to be trained, the consumed time is still over a few hours with our computer. From the result, we can see the DL-DBIS spends similar but a bit less time than that by human labeling. It shows that our DL-DBIS replaces human labeling tasks with similar time cost. The lower time cost can be achieved if more equipment resources are invested. Most importantly, a bean-picking proficient who is also skilled in the labeling tool only spends merely 175 min on labeling 35 images (around 2100 beans); then the proposed DL-DBIS works with approximately 70~80% accuracy, which meets the goal of reducing human effort. To the best of our knowledge, this is a pioneer work on removing defective beans with robotic arm plus deep learning techniques. Our following publications would take this version as a basis for performance comparison.

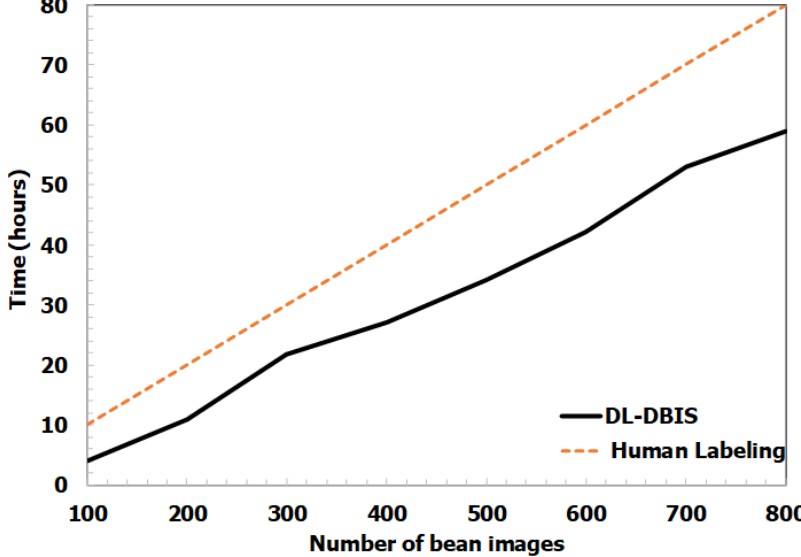

**Figure 18.** Comparison of labeling time between the proposed deep-learning-based defective bean inspection scheme (DL-DBIS) and human labeling.

### 7. Conclusions

In this paper, we proposed the a deep-learning-based defective bean inspection scheme (DL-DBIS), together with a GAN-structured automated labeled data augmentation method for enhancing the

proposed scheme, so that the automation degree of bean removal with robotic arms is significantly improved for coffee industries. Furthermore, all developed deep networks and related technical mechanisms are systematically presented in detail. Our proposed scheme needs only limited human time on bean labeling, which implies that this solution should satisfy the industrial needs. This work brings two main impacts to intelligent agriculture industries. First, our proposed scheme can easily be adopted by industries as human effort in labeling coffee beans are minimized. Second, our scheme can inspect all classes of defective beans categorized by the SCAA at the same time. These two achievements lead automation engineering and deep learning technologies into the coffee industries. We implemented a prototype of the proposed scheme to verify efficiency by applying the prototype to a robotic arm system. Testing results reveal that DL-DBIS can efficiently and effectively generate models for inspecting defective beans. Current experimental results are mainly based on the precision, recall, and accuracy. More statistical tests will provide thorough evidences to support the superior performance of the proposed scheme. We leave this issue as a reference to researchers who will extend this work in this area. This paper provides a useful reference for industrial practitioners of the coffee industry to construct the smart automation systems for producing high-quality coffee products with less human labor.

**Author Contributions:** Conceptualization, C.-J.K., M.-H.H. and C.-C.C.; Formal analysis, Y.-C.C. (Yung-Chien Chou), G.-J.H. and M.-E.W.; Methodology, C.-J.K., Y.-C.C. (Yung-Chien Chou), Y.-C.C. (Yi-Chung Chen) and Y.-C.L.; Project administration, M.-Y.P. and D.-C.W.; Software, C.-J.K., Y.-C.C. (Yung-Chien Chou), T.-T.C. and W.-T.S.; Industrial sponsorship management, G.J.H., M.Y.P. and Y.C.L.; Writing—original draft, Y.-C.C. (Yung-Chien Chou), C.-J.K. and C.-C.C.; Writing—review & editing, Y.-C.C. (Yung-Chien Chou) and C.-C.C.

**Funding:** This work was supported by Ministry of Science and Technology (MOST) of Taiwan under Grants MOST 107-2221-E-006-017-MY2, 107-2218-E-006-055, 108-2221-E-034-015-MY2, 107-2221-E-218-024, 108-2218-E-020-003, and 107-2221-E-156-001-MY2. This work was also supported by the "Intelligent Service Software Research Center" in STUST and the "Allied Advanced Intelligent Biomedical Research Center, STUST" under Higher Education Sprout Project, Ministry of Education, Taiwan. This work was financially supported by the "Intelligent Manufacturing Research Center" (iMRC) in NCKU from The Featured Areas Research Center Program within the framework of the Higher Education Sprout Project by the Ministry of Education (MOE) in Taiwan.

**Acknowledgments:** Authors thank ZhiJing Tsai and Min-Xiang Liu for collecting experimental data used in the case study. Authors also thank Steven Lin with AdvantTech Co. for providing practical comments on prototype development and promoting cooperation to meet industrial needs.

**Conflicts of Interest:** The authors declare no conflict of interest.

## Abbreviations

The following summarizes the acronyms used in this work. Note that abbreviations created in related works, e.g., YOLO, are not shown here.

| | |
|---|---|
| AIQC | augmented-image qualify check |
| BB | bounding box |
| BBC | bounding box class |
| BSDN | bean-shifting deep network |
| CP | class probability |
| DIA | Defect Inspection Accuracy |
| DIDN | defect-sensitive inspection deep network |
| DIP | Defect Inspection Precision |
| DIR | Defect Inspection Recall |
| DMQC | defect-sensitive model quality check |
| DSMC | defect-sensitive model creation |
| GDV | grid-description vector |
| GIMG | GAN-based image-augmentation model generation |
| LBIDB | labeled-bean image database |
| LBIG | low-detection-rate bean image generation |
| SPCOM | single-point crossover over matrix |

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
