# Peer review of "Deep-Learning-Based Defective Bean Inspection with GAN-Structured Automated Labeled Data Augmentation in Coffee Industry"

_applsci, doi:10.3390/app9194166_

Round 1
Reviewer 1 Report
In this manuscript, the authors design a deep learning based framework to inspect defective bean. GAN is utilized to augment the data to enhance the training. Overall, the manuscript is clearly written, it is well organized. The technical contribution is good, given the fact it specially designed to this specific problem.
I only have some minor comments before the acceptance of this manuscript.
GANs are widely applied to many field. Please discuss this more in details by citing the following papers.
[1] Tang et. al, XLSor: A Robust and Accurate Lung Segmentor on Chest X-Rays Using Criss-Cross Attention and Customized Radiorealistic Abnormalities Generation. International Conference on Medical Imaging with Deep Learning (MIDL), 2019 https://arxiv.org/abs/1904.09229
[2] Nie et. al, Medical Image Synthesis with Context-Aware Generative Adversarial Networks. Medical Image Computing and Computer Assisted Intervention − MICCAI 2017 https://link.springer.com/book/10.1007/978-3-319-66179-7
[3] Tang et. al, TUNA-Net: Task-oriented UNsupervised Adversarial Network for Disease Recognition in Cross-Domain Chest X-rays, Medical Image Computing and Computer Assisted Intervention − MICCAI 2019 https://arxiv.org/abs/1908.07926
Author Response
Reviewer #1:
In this manuscript, the authors design a deep learning based framework to inspect defective bean. GAN is utilized to augment the data to enhance the training. Overall, the manuscript is clearly written, it is well organized. The technical contribution is good, given the fact it specially designed to this specific problem.
Reply: Thank you for positively recognizing this work.
I only have some minor comments before the acceptance of this manuscript. GANs are widely applied to many field. Please discuss this more in details by citing the following papers.
[1] Tang et. al, XLSor: A Robust and Accurate Lung Segmentor on Chest X-Rays Using Criss-Cross Attention and Customized Radiorealistic Abnormalities Generation. International Conference on Medical Imaging with Deep Learning (MIDL), 2019 https://arxiv.org/abs/1904.09229 [2] Nie et. al, Medical Image Synthesis with Context-Aware Generative Adversarial Networks. Medical Image Computing and Computer Assisted Intervention − MICCAI 2017
https://link.springer.com/book/10.1007/978-3-319-66179-7
[3] Tang et. al, TUNA-Net: Task-oriented UNsupervised Adversarial Network for Disease Recognition in Cross-Domain Chest X-rays, Medical Image Computing and Computer Assisted Intervention − MICCAI 2019
https://arxiv.org/abs/1908.07926
Reply: We have added a new discussion (Lines 115-123) in Sec. 2 (Page 4) to the three related GAN works cited in [31-33] in the new version. We believe the new manuscript shall provide sufficient background references and readability to readers.

Reviewer 2 Report
Title Deep-Learning-Based Defective Bean Inspection with GAN-Structured Automated Labeled Data Augmentation in Coffee Industry
This is a paper on the application of many AI and ML techniques in coffee bean classification for the food production industry. First, the authors identify that collecting training data is labor intensive, hence a GAN scheme is proposed to generate more training sample based on limited human-labeled data. During classification the Hough based circle detector is used to act as feature detectors for classification of beans. Furthermore a GA-based Optimizer for the Proposed GAN Framework is used to help to generate data with more diversity. The system is evaluated and testing result is shown in figure 18.
This paper contains a number of interesting ideas and show they are applicable to real problem. Implementation is presented in detailed in the paper. The organization is logical, however, since it involves many different techniques during the research, a table ,or possible a flow diagram to show all modules (or research ideas) working interactively should be added to the introduction to give a clear picture of what the whole system looks like at the early part of the paper.
Some minor English grammatical problem should be fixed.
Line
2) one of most labor-consuming stage, --> one of most labor-consuming stages,
42)For detecting all of these detective beans as possible, some
43 recent works [3,12–15] solves this issue with machine/deep learning technologies
-->
For detecting all of these detective beans as possible, some
43 recent work [3,12–15] solves this issue with machine/deep learning technologies
or
For detecting all of these detective beans as possible, some
43 recent works [3,12–15] solve this issue with machine/deep learning technologies
Author Response
Reply Letter to Reviewers’ Comments (applsci-590986)
Reviewer #2:
This is a paper on the application of many AI and ML techniques in coffee bean classification for the food production industry. First, the authors identify that collecting training data is labor intensive, hence a GAN scheme is proposed to generate more training sample based on limited human-labeled data. During classification the Hough based circle detector is used to act as feature detectors for classification of beans. Furthermore a GA-based Optimizer for the Proposed GAN Framework is used to help to generate data with more diversity. The system is evaluated and testing result is shown in figure 18.
Reply: Thank you for the appropriate comment to our paper.
This paper contains a number of interesting ideas and show they are applicable to real problem. Implementation is presented in detailed in the paper. The organization is logical, however, since it involves many different techniques during the research, a table, or possible a flow diagram to show all modules (or research ideas) working interactively should be added to the introduction to give a clear picture of what the whole system looks like at the early part of the paper.
Reply: We agree with the reviewer that a table or a flow diagram shown the interaction between modules would increase readability. In this new version, we have added a module interaction table, shown below, to explicitly demonstrate how all modules work interactively. Notice that Fig. 4 introduces the overall architecture and the workflow of modules. Thus, the new table is placed right after Fig. 4 to assist readers fast and clearly understand the overall module interaction.
|
Module |
Interaction with related modules |
|
LBIDB |
provides the human-labeling training data to DSMC, GIMG, and LBIG. |
|
DSMC |
creates a defect-sensitive model for GIMG and DMQC. |
|
GIMG |
creates a generative network model for LBIG. |
|
LBIG |
generates and via BSDN for AIQC. |
|
AIQC |
verifies the quality of and from LBIG and produces and for DMQC. |
|
DMQC |
verifies the quality of from DSMC with and and activates DSMC for the next model training iteration if current model is not qualified. |
Some minor English grammatical problem should be fixed.
Line
2) one of most labor-consuming stage, --> one of most labor-consuming stages,
42)For detecting all of these detective beans as possible, some
43 recent works [3,12–15] solves this issue with machine/deep learning technologies
-->
For detecting all of these detective beans as possible, some
43 recent work [3,12–15] solves this issue with machine/deep learning technologies
or
For detecting all of these detective beans as possible, some
43 recent works [3,12–15] solve this issue with machine/deep learning technologies
Reply: We have revised the grammatical issues through the manuscript. The above-mentioned problems are fixed in Abstract on Page 1 (Line 2) and Section I on Page 2 (Line 46).

Reviewer 3 Report
This article provides a method for coffee bean (defect) classification based on the deep-learning approach.
This is a thorough work which provides the background of the case properly and introduces the authors approach to cover all aspects of coffee bean defects in detail. As such, the article could be a proper resource for other researchers interested in the field and the coffee bean industry alike.
However, I believe considering the following point would improve the article further.
Contextual:
In the last sentence of the abstract (lines 18-19), it would help if instead of “significantly” a measure (a percentage or a number) is given to show how significant and effective the method is.
To address some closely related work, for example, “Turi, B., Abebe, G. and Goro, G., 2013. Classification of Ethiopian coffee beans using imaging techniques”.
What is the reason for the IoT Device in Figure 3? Does it mean a device similar to the Raspberry Pi used in Figure 12? That is, is there any relation between the approach and the Internet of Things? If yes, where it has been described?
Where is the explanation of Step 4 of Figure 3?
It could have more consistent if the steps of Figure 4 have also been given in a similar way which Figure 3 has been described.
Is there any golden/base standard to compare the efficiency of the method in Section 6-6? For example, instead of saying “spend a few time”, line 553, having a number that shows the improvement against the previous work would be more appropriate.
Formative:
More proofreading would catch issues such as:
Using both abbreviations/acronyms and their expansions multiple time (e.g., lines 83 and 136).
“can … generating...”, line 18.
“cost … costly.”, line 17.

Author Response
Reply Letter to Reviewers’ Comments (applsci-590986)
Reviewer #3:
This article provides a method for coffee bean (defect) classification based on the deep-learning approach.
This is a thorough work which provides the background of the case properly and introduces the authors approach to cover all aspects of coffee bean defects in detail. As such, the article could be a proper resource for other researchers interested in the field and the coffee bean industry alike.
Reply: We appreciate the reviewer’s high acknowledgment to this work.
However, I believe considering the following point would improve the article further.
Contextual:
In the last sentence of the abstract (lines 18-19), it would help if instead of “significantly” a measure (a percentage or a number) is given to show how significant and effective the method is.
Reply: We have added explicit numbers obtained from experiments to concretely present our results in Abstract on Page 1 (Lines 17-19).
To address some closely related work, for example, “Turi, B., Abebe, G. and Goro, G., 2013. Classification of Ethiopian coffee beans using imaging techniques”.
Reply: This related work is indeed close to our paper scope. We have cited this paper, together with new discussion, in Introduction on Page 2 (Lines 42-45) of the new version.
What is the reason for the IoT Device in Figure 3? Does it mean a device similar to the Raspberry Pi used in Figure 12? That is, is there any relation between the approach and the Internet of Things? If yes, where it has been described?
Reply: The IoT (Internet of Things) device in general provides computation and communication capacity. Our proposed scheme requires computation resources to achieve the defective bean identification and informs the robot controller with the communication function. Currently, many IoT productions are invented and can be adopted in this work. Raspberry Pi is one of existing productions which can be easily found in the today’s Markets, and indeed we use Raspberry Pi in our implementation.
For ease of understanding, we replace the “IoT device” by the term “single-chip computer” to emphasize the computational requirement for avoiding confusion on its Internet capacity. We have added the above explanation in Sec. 3 (Page 5, Lines 163-166) and in Fig. 3 in the new version.
Where is the explanation of Step 4 of Figure 3?
Reply: It is our mistake on the writing consistency. We have revised Section 3 on Page 6 (Lines 188-190) to add Step 4 (and its explanation) of Figure 3 in the new version.
It could have more consistent if the steps of Figure 4 have also been given in a similar way which Figure 3 has been described.
Reply: We agree with the reviewer such the enumerated expression is easier to get the main idea to this figure. We have rewritten the Sec. 3.1 as the reviewer’s request on Pages 6-7 (Lines 223-237) in the new version.
Is there any golden/base standard to compare the efficiency of the method in Section 6-6? For example, instead of saying “spend a few time” in line 553, having a number that shows the improvement against the previous work would be more appropriate.
Reply: We don’t find any standard to compare the efficiency in the related industries so far. To the best of our knowledge, this is a pioneer work on removing defective beans with robotic arm plus deep learning techniques. We agree that a golden standard can make the comparison be widely accepted by researchers. Our next publications would take this version as a basis for more comprehensive comparisons.
In the new version, we have added the quantity representation in the content, and remove the uncertain expression, such as “spend a few time”.
Formative:
More proofreading would catch issues such as:
Using both abbreviations/acronyms and their expansions multiple time (e.g., lines 83 and 136).
“can … generating...”, line 18.
“cost … costly.”, line 17.
Reply: We have revised the grammatical issues through the manuscript, including all problem on Pages 1-4 (Lines 17-19, 32, and 85) according to reviewer’s requests.

Reviewer 4 Report
I am reading this paper several times but always with many questions what follows from the many abbreviation, indications signs, etc. So it is difficult to read.
Although this paper seems to be interesting, form of results presentation and theoretical background are complicated. The paper is also to long.
In the paper all abbreviations should be first developed (IMG on Fig. 2 ) means probably the word IMAGE but reader have to be informed that it is. Similarly “IoT device” (probably internet of things device), and many others.
In many places variables in the equations, for example in (6) are explained further so reader in this place confused. Similarly variable “k: practically in all equations is not explained.
In eqs. (9,10,11) all elements are undefined and not explained!
I also don’t understand why from Fig. 14 follows that proposed approach is better, who prepare these images?
Experimental part of the paper is enough and convince but I propose using also strong statistical test (Wilcoxon) to comparison of different methods.
Other parts of the manuscript should be improved.
Author Response
Reply Letter to Reviewers’ Comments (applsci-590986)
Reviewer #4:
I am reading this paper several times but always with many questions what follows from the many abbreviation, indications signs, etc. So it is difficult to read. Although this paper seems to be interesting, form of results presentation and theoretical background are complicated. The paper is also too long.
Reply: We has added an acronym table in Appendix A (Page 23) of the new version so that readers can easily look up abbreviations while reading the technical details. The acronym table is shown below. Moreover, Reviewer #1 also recommends to add some deep learning works to the background section, which have been done in the new version. We believe these new references assist readers to fast understand the related background knowledge to this area.
|
Abbreviation |
Full name |
Abbreviation |
Full name |
|
AIQC |
augmented-image qualify check |
DIR |
Defect Inspection Recall |
|
BB |
bounding box |
DMQC |
defect-sensitive model quality check |
|
BBC |
bounding box class |
DSMC |
defect-sensitive model creation |
|
BSDN |
bean-shifting deep network |
GDV |
grid-description vector |
|
CP |
class probability |
GIMG |
GAN-based image-augmentation model generation |
|
DIA |
Defect Inspection Accuracy |
LBIDB |
labeled-bean image database |
|
DIDN |
defect-sensitive inspection deep network |
LBIG |
low-detection-rate bean image generation |
|
DIP |
Defect Inspection Precision |
SPCOM |
single-point crossover over matrix |
In addition, Reviewer #3 asks us to rewrite Sec. 3.1 as an enumerated style, which is also done in the new version. Such representation makes content more structural and assists readers to easily get the main idea of each modules in our proposed scheme. We expect that the new structure assists readers to further build the big picture of the overall concept to our proposed scheme in readers’ mind, and slightly relieves reading load from this technique-comprehensive paper.
In the paper all abbreviations should be first developed (IMG on Fig. 2) means probably the word IMAGE but reader have to be informed that it is. Similarly “IoT device” (probably internet of things device), and many others.
Reply: We have corrected “IMG” to “Image” in Fig. 2 (on Page 4) for avoid confusion. Also, “IoT Device” has been corrected to “single-chip computer”. We have updated other similar issues through the paper.
In many places variables in the equations, for example in (6) are explained further so reader in this place confused. Similarly variable “k: practically in all equations is not explained.
Reply: We have added explanation to variables in the new version. Some of them are listed as follow:
(4) on Page 11 (Line 322). (6) on Page 15.
The variable k is a global counter used for tracking the model training iterations in the main framework (referring to Step 7, Fig. 4). Thus, k is used in other variables of related modules for retrieving historical training data, and it is the reason why it frequently appears in the following sections. For avoid misleading readers, we added the above explanation in Sec. 3 (Page 7, Lines 238-241) to emphasize this variable.
In eqs. (9,10,11) all elements are undefined and not explained!
Reply: We have revised Eqs. (9)-(11) on Pages 17-18 (Lines 462, 464-465 and 466-467) in the new version.
I also don’t understand why from Fig. 14 follows that proposed approach is better, who prepare these images?
Reply: This is misleading. Fig. 14 shows various types of generated image used in training the optimal bean identification model, where Fig. 14(a) is the original image labeled by human, and Fig. 14(b)-(d) are generated with different densities by our proposed LBIG module presented in Sec. 4.5. This is our means to increase training data, so that our generated models of identifying defective beans perform better than others. We visualize some of them for demonstrating this property of our scheme, which is happened in the stage of executing the LBIG module.
We have written Sec. 6.2 and added the above explanation on Page 18 (Lines 470-474 and 476-478) in the new version.
Experimental part of the paper is enough and convince but I propose using also strong statistical test (Wilcoxon) to comparison of different methods. Other parts of the manuscript should be improved.
Reply: We agree that statistical tests will be another good means to convince readers the superior performance of proposed scheme. Like the reviewer’s comments, current version of the experiments in this paper have been sufficient and convincing as well. Thus, we leave this issue as a future direction, and this would be a useful reference to other researchers who extend this work in this area.
We have added the above explanation in Sec. 7 of the new version (Pages 22-23, Lines 584-587). Similar issues mentioned by the reviewers are completely revised in the new version.
